# A multiplex of connectome trajectories enables several connectivity patterns in parallel

Parham Mostame[1,2]*, Jonathan Wirsich[3], Thomas Alderson[1,2], Ben Ridley[4,5], Anne-Lise Giraud[6], David W Carmichael[7,8], Serge Vulliemoz[3], Maxime Guye[4,9], Louis Lemieux[10,11], Sepideh Sadaghiani[1,2]

[1]Department of psychology, University of Illinois at Urbana-Champaign, Champaign, United States; [2]Beckman Institute for Advanced Science and Technology, University of Illinois at Urbana-Champaign, Champaign, United States; [3]EEG and Epilepsy Unit, University Hospitals and Faculty of Medicine of Geneva, University of Geneva, Geneva, Switzerland; [4]Aix Marseille Univ, CNRS, CRMBM, Marseille, France; [5]IRCCS Istituto delle Scienze Neurologiche di Bologna, Bologna, Italy; [6]Department of Neuroscience, University of Geneva, Geneva, Switzerland; [7]Developmental Imaging and Biophysics Section, UCL Great Ormond Street Institute of Child Health, London, United Kingdom; [8]School of Biomedical Engineering and Imaging Sciences, King's College London, St Thomas' Hospital, London, United Kingdom; [9]AP-HM, Hôpital Universitaire Timone, Pôle d'Imagerie Médicale, CEMEREM, Marseille, France; [10]Department of Clinical and Experimental Epilepsy, UCL Queen Square Institute of Neurology, London, United Kingdom; [11]Epilepsy Society MRI Unit, Chalfont St Peter, Buckinghamshire, United Kingdom

*For correspondence: mostame2@illinois.edu

Competing interest: The authors declare that no competing interests exist.

## eLife Assessment

This **important** work uses an innovative approach to understand similarities between haemodynamic and electrophysiological activity of the human brain, and how the brain might carry out multiple functions concurrently across different brain regions by using multiple timescales. The study provides **convincing** evidence to indicate that while spatially similar functional brain networks are found in both modalities, there is a tendency for these to occur asynchronously. This work will be of interest to neurophysiological and brain imaging researchers.

**Abstract** Complex brain function comprises a multitude of neural operations in parallel and often at different speeds. Each of these operations is carried out across a network of distributed brain regions. How multiple distributed processes are facilitated in parallel is largely unknown. We postulate that such processing relies on a multiplex of dynamic network patterns emerging in parallel but from different functional connectivity (FC) timescales. Given the dominance of inherently slow fMRI in network science, it is unknown whether the brain leverages such multi-timescale network dynamics. We studied FC dynamics concurrently across a breadth of timescales (from infraslow to γ-range) in rare, simultaneously recorded intracranial EEG and fMRI in humans, and source-localized scalp EEG-fMRI data in humans. We examined spatial and temporal convergence of connectome trajectories across timescales. 'Spatial convergence' refers to spatially similar EEG and fMRI connectome patterns, while 'temporal convergence' signifies the more specific case of spatial convergence at corresponding timepoints in EEG and fMRI. We observed spatial convergence but temporal divergence across FC timescales; connectome states (recurrent FC patterns) with partial spatial

similarity were found in fMRI and all EEG frequency bands, but these occurred asynchronously across FC timescales. Our findings suggest that hemodynamic and frequency-specific electrophysiological signals, while involving similar large-scale networks, represent functionally distinct connectome trajectories that operate at different FC speeds and in parallel. This multiplex is poised to enable concurrent connectivity across multiple sets of brain regions independently.

## Introduction

Numerous distributed neural processes unfold in parallel in support of complex brain function. Speech comprehension, for example, requires concurrent processing of phonemes, syllables, and sentence-level prosody, syntax, and semantics. An essential characteristic of such parallel processes is that they occur at different timescales (*Kahneman, 2011*; *Perdikis et al., 2011*; *Hari and Parkkonen, 2015*; *Kringelbach et al., 2015*), in the case of speech spanning from tens of milliseconds (phonemes), over hundreds of milliseconds (syllables), to several seconds (sentences). Irrespective of timescale, each of these subprocesses relies on a distributed set of functionally connected brain regions. Given the parallel nature and different timescales of the subprocesses, FC among multiple sets of regions and at multiple speeds may be required in parallel. However, due to the dominance of inherently slow fMRI in the study of FC, it is unknown whether the brain's connectivity architecture supports such multi-timescale connectivity.

The functional connectome constitutes the whole-brain spatial organization of large-scale FC and is thought to represent the brain's omnipresent and intrinsic 'cognitive architecture' (*Petersen and Sporns, 2015*). This spatial organization continuously exhibits reconfigurations across different spatial patterns in all mental states, including task-free rest. These reconfigurations are referred to as spontaneous or ongoing FC dynamics (*Preti et al., 2017*; *Lurie et al., 2018*; *Sadaghiani and Wirsich, 2020*). Such ongoing FC dynamics as observed in both fMRI and electrophysiology are associated with cognitive performance (*Weisz et al., 2014*; *Sadaghiani et al., 2015*; *Cohen, 2018*; *Rassi et al., 2019*). More specifically, spontaneously occurring FC dynamics include certain patterns thought to represent distinct discrete connectome states supporting different cognitively and behaviorally relevant processes; Thus, iteration through these states may be commensurate with maintaining cognitive flexibility during all cognitive states including rest (*Bassett et al., 2011*; *Deco et al., 2013*; *Chen et al., 2016*; *Vohryzek et al., 2020*; *Alavash et al., 2021*).

The functional connectome is commonly studied with fMRI as it offers whole-brain coverage and precise localization. The discovery of intrinsic large-scale FC organization around the turn of the millennium (*Biswal et al., 1995*; *Greicius et al., 2003*; *Damoiseaux et al., 2006*) and its time-varying dynamics over the following two decades (*Chang and Glover, 2010*; *Majeed et al., 2011*; *Handwerker et al., 2012*; *Allen et al., 2014*) were based on fMRI data. It is perhaps for this reason that the literature commonly treats functional connectome dynamics as conceptually synonymous with these fMRI-derived observations. However, while fMRI is limited to the infraslow timescale of the hemodynamic response, it is rarely considered whether large-scale functional connectome dynamics comprise other FC processes not readily measurable in fMRI.

Beyond fMRI, a small but growing body of work suggests that the functional connectome can be reliably observed using electrophysiological methods such as MEG, EEG, and intracranial EEG (iEEG) (*Brookes et al., 2011*; *Sadaghiani and Wirsich, 2020*; *Sadaghiani et al., 2022*). Theoretically, FC as captured by fMRI and electrophysiology is partly distinct with respect to the underlying neural populations, connectivity mechanisms, and timescales. Specifically, while the Local Field Potential signal of electrophysiological methods including intracranial recordings is dominated by pyramidal neurons (*Buzsáki et al., 2012*), the fMRI signal stems from the metabolic demands of neural activity that can cumulatively reflect a large variety of neural populations (*Heeger and Ress, 2002*). Regarding FC in particular, fMRI and electrophysiological recordings differ in terms of both speed and connectivity mechanisms; fMRI captures connectivity based on slow co-fluctuations of the hemodynamic signal, thus resulting in connectome dynamics in the range of seconds to minutes (*Chang and Glover, 2010*; *Calhoun et al., 2014*; *Vohryzek et al., 2020*). Conversely, connectivity inferred from electrophysiological data is commonly based on cross-region coupling of phase or amplitude of fast neural processes (~1–150 Hz), resulting in connectome dynamics at sub-second speeds (*Baker et al., 2014*;

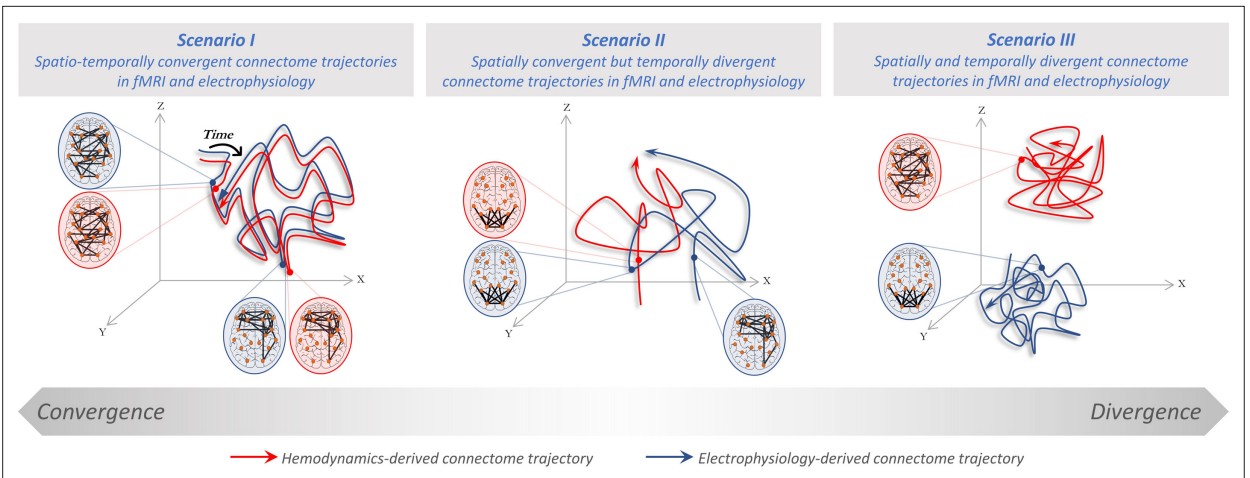

**Figure 1.** Three potential scenarios for convergence/divergence of connectome trajectories across hemodynamics and electrophysiology. Dynamics of connectome reconfigurations (i.e. time-varying spatial patterns of large-scale connectivity) are shown as trajectories (functional connectivity, FC pattern sequences) in a presumed 3D state space, starting from the diamond sign and ending at the arrowhead. Hemodynamics- and electrophysiology-derived connectome dynamics in each scenario are shown in blue and red, respectively. Scenario (I): both dynamic trajectories are driven by the same connectivity processes; are spatially and temporally convergent (at all times). Scenario (II): both dynamic trajectories span the same portion of the state space, while each traversing a different timecourse; are spatially convergent but temporally divergent. Scenario (III): each of the dynamic trajectories spans non-overlapping parts of the state space with different timecourses; are spatially and temporally divergent (at all times). Note that scenarios I and III represent border cases extracted from a theoretical convergence/divergence gradient (bottom of the figure). Spatial convergence does not imply a perfect spatial match (at the same or different time points). Rather, spatial convergence –if observed– is likely to be partial, given that the cross-modal similarity of static connectomes is known to be moderate (*Betzel et al., 2019*; *Wirsich et al., 2021*). Furthermore, note that electrophysiology-derived connectivity is simplified for illustration purposes and may comprise several trajectories at different frequency bands.

The online version of this article includes the following figure supplement(s) for figure 1:

**Figure supplement 1.** Electrode locations for each intracranial EEG (iEEG)-fMRI subject shown in standard MNI space.

*Hunyadi et al., 2019*). Such sub-second dynamics are thought to enable frequency-specific modes of rapid neural communication (*Engel et al., 2013*; *Mostame and Sadaghiani, 2021*).

Given that spontaneous connectome dynamics as observed in *separate* fMRI and electrophysiological experiments may stem from partially distinct neural populations and FC timescales (*Nunez and Silberstein, 2000*; *Hari and Parkkonen, 2015*; *Hermes et al., 2017*), a fundamental question arises: Could the connectome entail more than one FC pattern at any given timepoint? It is conceivable that multiple connectome state sequences, or dynamic trajectories, concurrently unfold at different timescales and serve as parallel 'channels' of FC. In this scenario, any given brain region may concurrently undergo dynamic connectivity to *multiple* distinct sets of other regions, with a different FC timescale for each set. This scenario may minimize interference and maximize information flow. We liken this multitude of communication channels to multiplexing in telecommunication, where independent information streams are conveyed through a single radio or television transmission medium. Such a multiplex would be ideally poised to support the multitude of cognitively relevant processes that occur at numerous speeds in parallel. To empirically assess whether the brain contains such a multiplex, all timescales of FC connectivity must be investigated concurrently, therefore, calling for multi-modal approaches.

Multimodal studies have begun to directly investigate the relationship between hemodynamics- and electrophysiology-derived large-scale FC (e.g. *He et al., 2008*; *de Pasquale et al., 2010*). This line of work shows that the *time-averaged or static* spatial organization of functional connectomes is (at least moderately) correlated between fMRI and electrophysiological data, including MEG (*Brookes et al., 2011*; *Hipp and Siegel, 2015*; *Shafiei et al., 2022*), EEG (*Deligianni et al., 2014*; *Wirsich et al., 2017*; *Wirsich et al., 2021*), and iEEG (*Kucyi et al., 2018*; *Betzel et al., 2019*). However, the *temporal* relationship of connectome dynamics across FC timescales, which requires concurrent fMRI and electrophysiological recordings, has rarely been studied. Understanding this temporal relationship will determine whether the brain leverages a multiplex of communication channels.

To understand the relationship of connectome dynamics across FC timescales, we measure the degree to which simultaneously recorded fMRI- and electrophysiology-derived connectome trajectories converge. *Figure 1* illustrates three possible scenarios. In scenario I, spatially similar connectome patterns occur in the two modalities at the same time once accounting for the hemodynamic delay (i.e. are spatially and temporally convergent). This scenario aligns with the common assumption that electrophysiological and hemodynamic measurements mostly show the same neural functional connectivity processes. The latter, in this perspective, essentially records a smoothed-out version (specifically convolved with the Hemodynamic Response Function or HRF) of the former. However, the above-described divergence of the neurophysiological bases of EEG and fMRI signals warrant alternative scenarios. In particular, in scenario II spatially similar connectome patterns occur in the two data modalities; however, at different timepoints (spatially convergent but temporally divergent). Conversely, in scenario III connectome patterns of the two modalities bear no spatial resemblance (spatially and temporally divergent). Note that a fourth scenario of temporally convergent but spatially dissimilar FC patterns is theoretically possible but is outside of our scope due to the infinite search space. We adjudicate these scenarios in a frame-by-frame manner, first in rare human concurrent fMRI and iEEG data providing optimal spatial precision (N=9, including individual non-parametric testing against subject-specific null models), then extending to whole-brain connectomes using a concurrent *source-localized* scalp EEG and fMRI dataset (N=26).

## Results

This study quantifies the spatio-temporal convergence of dynamic reconfigurations of connectomes driven by different neural timescales at which FC is derived (infraslow to γ-band). Note that here timescale refers to the speed of FC rather than the speed at which FC dynamically changes. We calculated spatial FC patterns in fMRI (fMRI-FC) and band-limited electrophysiology (EEG-FC) at every fMRI volume (repetition time or TR). The window-wise FC patterns were then spatially compared across data modalities at *all* possible time lags. This novel technique, which we call *cross-modal recurrence plot* or CRP, allows for simultaneous investigation of synchronous and asynchronous spatial convergence in bimodal datasets. Results for amplitude coupling (EEG-FC$_{Amp}$) are provided in the main manuscript, and replications using phase coupling (EEG-FC$_{Phase}$) are provided in Appendix 1 and *Figure 3—figure supplement 4*. The CRP results are first provided without HRF-convolution (but with a 6 s lag of the fMRI data) to retain the full temporal range of dynamic information in electrophysiological signals. Then, CRPs are investigated with HRF-convolution to aid interpretation. Analysis is first performed on iEEG-fMRI, followed by whole-brain source-localized scalp EEG-fMRI in a separate healthy cohort. For completeness, we also replicated previous reports of cross-modal spatial concordance between *static*, i.e., time-averaged, connectomes (see Appendix 1).

### Comparison of fMRI and iEEG dynamic connectome patterns

#### Cross-modal recurrence plot (CRP)

We investigated the spatial similarity of fMRI and iEEG connectome patterns at all possible pairs of timepoints (adopted from a prior unimodal approach for fMRI *Hansen et al., 2015*; *Cabral et al., 2017*). In other words, we asked whether each frame-wise fMRI-FC pattern was spatially similar (i.e. spatially correlated) to each frame-wise iEEG-FC$_{Amp}$ pattern at *any* timepoint of the recording, and vice versa. For each subject, the cross-modal correlation matrix was separately estimated for each electrophysiological frequency band, resulting in a 2D *cross-modal* correlation matrix per subject and frequency band. Statistical evaluation of correlation values against null models resulted in thresholded and binarized matrices: *Cross-modal Recurrence Plots* (*Figure 2A*). To investigate the multi-frequency nature of electrophysiology, we overlaid the binary CRP matrices of the five frequency bands into a *multi-frequency* CRP for each subject (*Figure 2B*). In the following, we adjudicate the likelihood of the three presumed scenarios of cross-modal convergence by investigating features of the multi-frequency CRPs (*Figure 2C*).

#### Cross-modal spatial convergence

Across all subjects, 20 ± 8% of all entries in the multi-frequency CRP (including on-diagonal and off-diagonal) showed significant spatial cross-modal convergence in at least one of the frequency bands

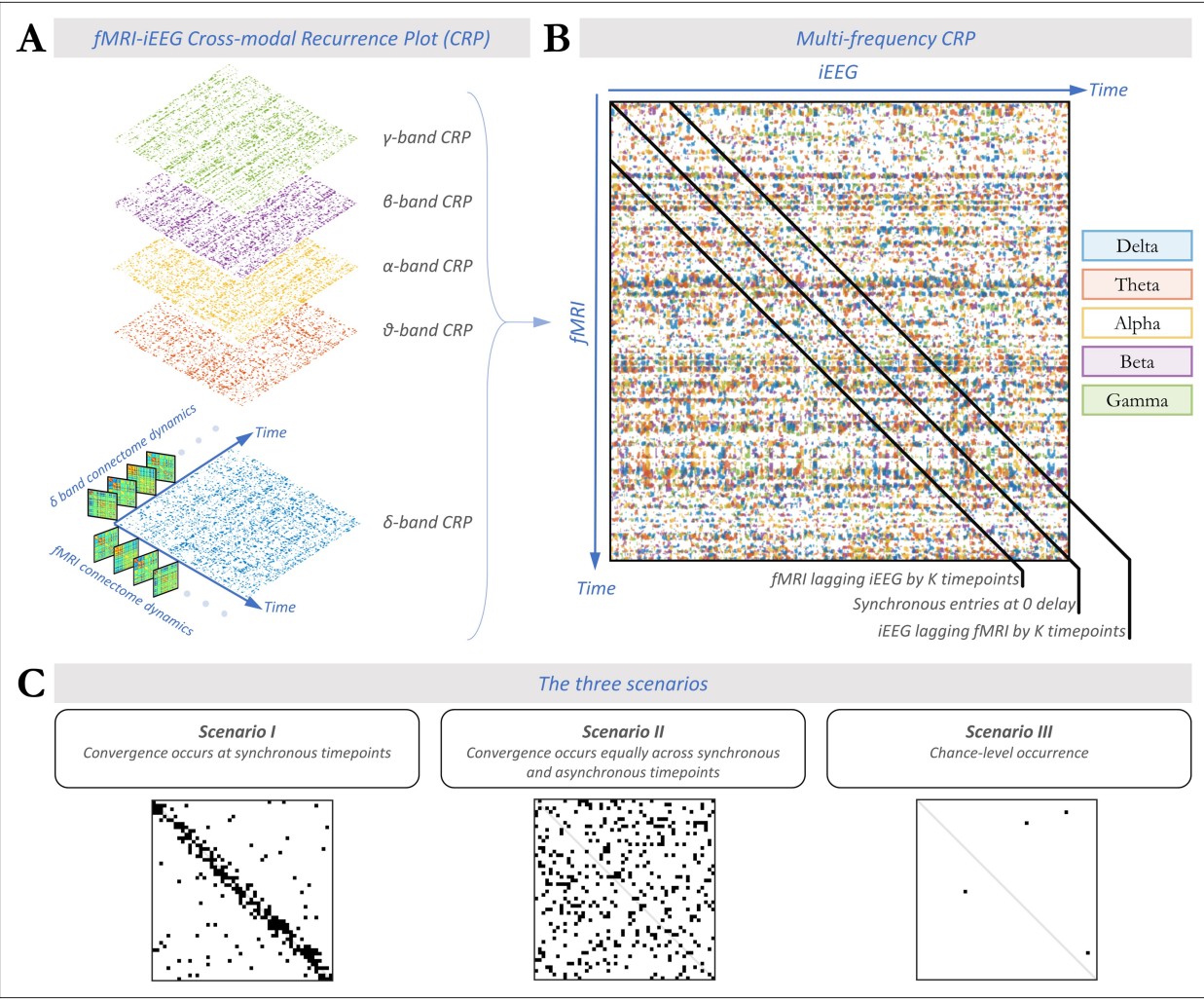

**Figure 2.** Cross-modal recurrence plots (CRPs) quantify synchronous and asynchronous spatial convergence across data modalities. (**A**) CRP between fMRI-FC and iEEG-FC_Amp shown for different electrophysiological bands (δ to γ) for a single subject. Note that synchronous observations (after accounting for a 6 s hemodynamic delay) fall on the diagonal of the CRP. The CRP in each frequency band is binarized, with correlation values passing the threshold if they exceed a null model generated from spatially phase-permuted FC matrices (Benjamini-Hochberg FDR-corrected for the number of all pairs of timepoints). (**B**) Multi-frequency CRP constructed for the same subject by overlaying the CRPs across all electrophysiological frequency bands. Pairs of timepoints with significant spatial correlation across fMRI and intracranial EEG (iEEG) functional connectivity (FC) are indicated with a different color for each frequency band. (**C**) Schematic views of expected CRP patterns corresponding to each of the three scenarios introduced in *Figure 1*. Significant CRP entries are marked black. The density of black bins depicts the extent of spatial convergence, while their prominence along the diagonal (measured as on-/off-diagonal ratio) translates to the extent of temporal convergence. In the following, the density and on-/off-diagonal ratio of significant entries in the multi-frequency CRP will be used to adjudicate the likelihood of the scenarios depicted in *Figure 1* (see section 'Validation of on-/off-diagonal ratio metric' in Appendix 1).

(*Figure 3B*). For these entries, the spatial Pearson correlation values are presented in *Figure 3A*. The observed spatial convergence, even though moderate to weak, suggests that scenario III is less compatible with our data since it posits that dynamic connectome patterns captured in fMRI and electrophysiology lack any spatial convergence. To adjudicate between scenarios I and II, the following sections assess to what extent the observed cross-modal spatial convergence occurs in a synchronous or asynchronous manner.

## Asynchronous nature of cross-modal convergence

Although scenarios I and II both imply spatial convergence of connectome dynamics, they differ in the extent of temporal convergence. In the multi-frequency CRP, this difference can be assessed in the *rate* of spatially convergent epochs at corresponding or synchronous timepoint pairs (*on-diagonal*

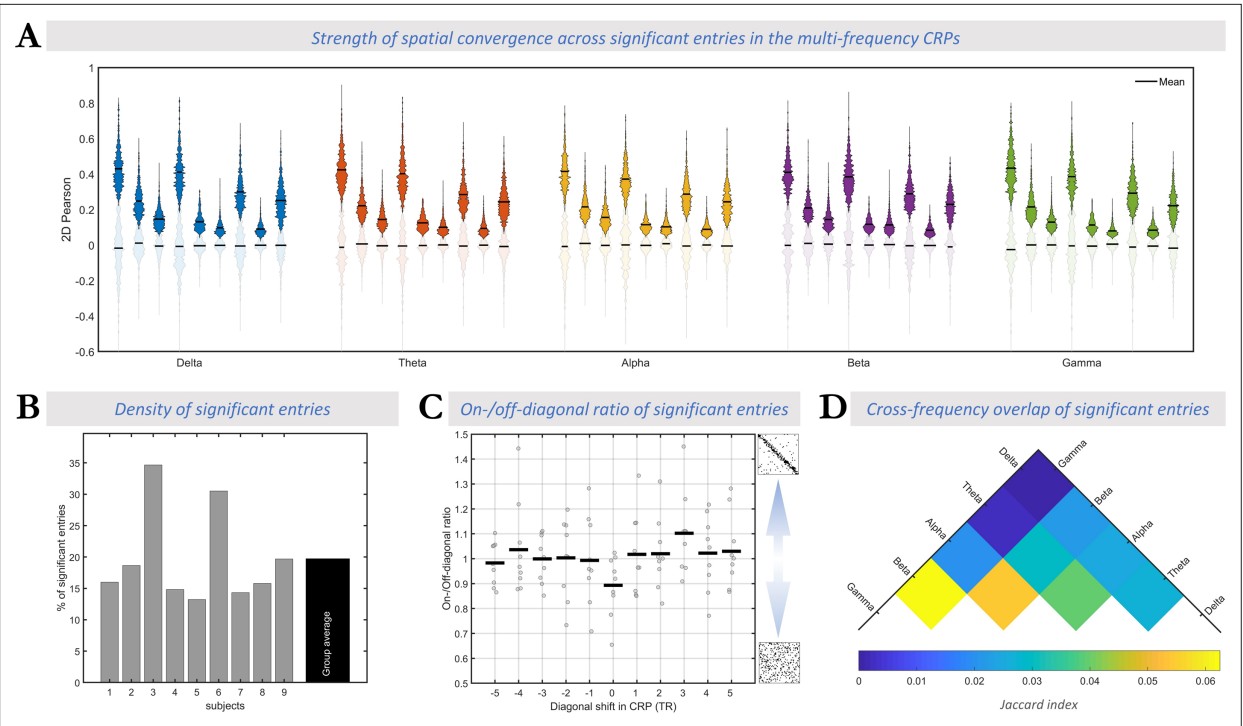

**Figure 3.** The spatial convergence across fMRI and frequency-specific intracranial EEG (iEEG) connectomes occurs asynchronously. (**A**) Effect size of cross-modal correlation: Frequency bands are color-coded using the same set of colors from *Figure 2*. Within each frequency band, each pair of opaque and transparent violin plots corresponds to one subject. Opaque (or transparent) plots represent the distribution of correlation values in significant (or insignificant) multi-frequency cross-modal recurrence plot (CRP) entries. The horizontal black lines in each violin plot represent the mean value of the distribution. Across all frequency bands, the effect size of spatial correlation between fMRI and iEEG-FC$_{Amp}$ connectome frames at significant epochs is considerably larger than in insignificant epochs. (**B**) Total density of significant epochs: Horizontal axis shows the subject number (rightmost column shows group average), while vertical axis shows percentage of significant epochs across all multi-frequency CRP entries. Overall, 20% of multi-frequency CRP entries averaged over subjects passed significance testing, indicating spatial convergence between fMRI and iEEG connectomes and speaking against scenario III. (**C**) On-/off-diagonal ratio of significant epochs calculated across a range of diagonal shifts in CRP. The horizontal axis shows a range of lead/lags by 5 TRs at which fMRI and iEEG were assumed to be aligned (i.e. 'on-diagonal') (see *Figure 2*). Recall that zero lag in CRP corresponds to original preprocessed data where fMRI is shifted 6 s to account for hemodynamic delay. Gray dots show the ratio for each individual and the horizontal lines represent the group-average ratios at each lag. Overall, the ratios are very close to 1 and do not exceed data-driven null distribution. This observation speaks in favor of scenario II where fMRI and iEEG connectome reconfigurations follow distinct temporal trajectories while manifesting spatially similar connectome patterns. (**D**) Degree of overlap between pairs of frequency-specific binarized CRPs (as shown in *Figure 2A*), expressed as group-average of the Jaccard index. The low values (cf. possible index range of 0–1) demonstrate that timepoint pairs of high cross-modal spatial correlation are largely distinct across different electrophysiological frequency bands. In summary, the observed cross-modal spatial convergence at frequency-specific and asynchronous epochs suggests that connectome dynamics across different functional connectivity (FC) timescales traverse spatially similar patterns asynchronously (Scenario II; *Figure 5*).

The online version of this article includes the following figure supplement(s) for figure 3:

**Figure supplement 1.** Static intracranial EEG (iEEG)/EEG connectome as an attractor for the cross-modal convergence.

**Figure supplement 2.** Temporal association between connection-wise dynamic functional connectivity (FC) changes of fMRI and intracranial EEG (iEEG).

**Figure supplement 3.** Replication of the results in the scalp EEG dataset after correction for volume conduction effects using source-orthogonalized signals for both amplitude- and phase-coupling measures (similar to *Figure 4* in the main text).

**Figure supplement 4.** Replication of major findings using phase coupling as a measure of electrophysiology connectivity in intracranial EEG and source-localized scalp EEG datasets.

**Figure supplement 5.** The on-/off-diagonal ratio is highly replicable when using a window size of 500 ms compared to original findings with a window size of one TR.

**Figure supplement 6.** cross-modal recurrence plot (CRP) calculated for simulated data with two different cases.

**Figure supplement 7.** On-/off-diagonal ratio across varying quantization levels (N) for simulated data with randomized signals (Scenario II) and identical signals (Scenario I).

entries) and lagged or asynchronous timepoint pairs (*off-diagonal* entries). Specifically, we quantified the ratio of on-diagonal to off-diagonal rates (hereinafter '*on-/off-diagonal ratio.*' See *Figure 3C*). In scenario I, fMRI and iEEG connectome trajectories are temporally aligned, hence spatially converging more commonly at synchronous than asynchronous timepoints. Thus, epochs of cross-modal spatial similarity should occur more frequently at on-diagonal (synchronous) than off-diagonal (asynchronous) entries, resulting in an on-/off-diagonal ratio larger than unity (ratio >>1). In scenario II, however, epochs of spatial similarity could occur equally likely at on-diagonal and off-diagonal entries (ratio ≈ 1) akin to a random multi-frequency CRP (see section 'Validation of On-/Off-diagonal ratio metric' in Appendix 1). Thus, we compared each subject's on-/off-diagonal ratio to a null distribution of ratios extracted after spatially scrambling the multi-frequency CRP along both temporal axes for 100 repetitions. Note that given the random spatial organization of the surrogate multi-frequency CRPs the null distribution of the ratios is expected to be centered around unity, indicating the absence of synchronous convergence.

Indeed, the on-/off-diagonal ratio was close to rather than exceeding 1 (mean over subjects ± std = 0.89±0.12), showing that synchronous convergence is no more likely than asynchronous convergence, which speaks against scenario I (*Figure 3C*). This observation was statistically confirmed in all individual subjects (non-parametric test of the on-/off-diagonal ratio against subject-specific null models; p>0.32 in all subjects). We directly assessed the probability of H0 (scenario II: absence of temporal convergence, i.e. ratio not exceeding 1) against the probability of H1 (scenario I: presence of temporal convergence, i.e. ratio larger than 1). Specifically, we performed a Bayesian one-tailed group-level paired *t*-test between the observed on-/off-diagonal ratios and the subject-specific mean of the null on-/off-diagonal ratios. A $BF_{01}$ (i.e. *Bayes Factor* in favor of H0 over H1) of 8.2 showed that the data are at least eight times more likely to occur under H0 (scenario II) than under H1 (scenario I).

Recall that we assumed a 6 s delay of fMRI relative to the iEEG signal due to the hemodynamic delay. As the assumed lag affects what data points are considered to be 'on-diagonal' (see *Figure 2B*), we addressed potential imprecisions of the assumed lag. Specifically, we additionally quantified the on-/off-diagonal ratio for a wide range of diagonal shifts in CRP, including from –15s to 15s (equivalent to ±5 TRs, where 0 TR corresponds to the original 6 s shift) (*Figure 3C*). Irrespective of lag, the on-/off-diagonal ratio remained non-distinguishable from 1 (mean ratio = 1.02 ± 0.03, mean null ratio = 1.02 ± 0.01, $BF_{01}$=3.5 ± 1.6).

Taken together, epochs of cross-modal spatial convergence are equally likely at asynchronous timepoints as they are at synchronous times; corresponding to the lack of an on- vs. off-diagonal pattern in the multi-frequency CRP. The observation of a different pattern, namely subtle horizontal stripes (*Figure 2B*), is discussed in Appendix 1.

## Multi-frequency cross-modal association

Overlaying the CRPs of all electrophysiological frequency bands allowed us to study the complementary nature of the cross-modal relationship in the different bands. In each subject, we quantified the extent of overlap between significant CRP entries for every pair of frequency bands using the Jaccard index. The group-average results are visualized for all pairs of frequencies in *Figure 3D*. Across all pairs of frequency bands, there was only a small overlap between the single-frequency CRPs (Jaccard index reaching at most 0.06±0.03 for β- vs. γ-band CRPs). This result demonstrates that the observed asynchronous spatial similarity of fMRI-FC and iEEG FC occurs in a temporally independent manner across all electrophysiological frequency bands (i.e. timescales).

## Replication with HRF-Convolved iEEG

We aimed to assess whether the observed temporal divergence —at the level of connectomes— is simply due to the different temporal characteristics of fMRI and iEEG regional signals. In this perspective, the former would merely constitute slow components, or a slowed-down version, of the latter. Thus, we repeated the CRP analysis after convolving the amplitude of region-wise band-limited iEEG signals with the canonical HRF to match the speed of fMRI signals.

## HRF-convolved iEEG

In agreement with our original analysis, the multi-frequency CRPs showed 22 ± 7% significant entries, indicating that scenario III does not explain our data. The multi-frequency CRPs extracted from

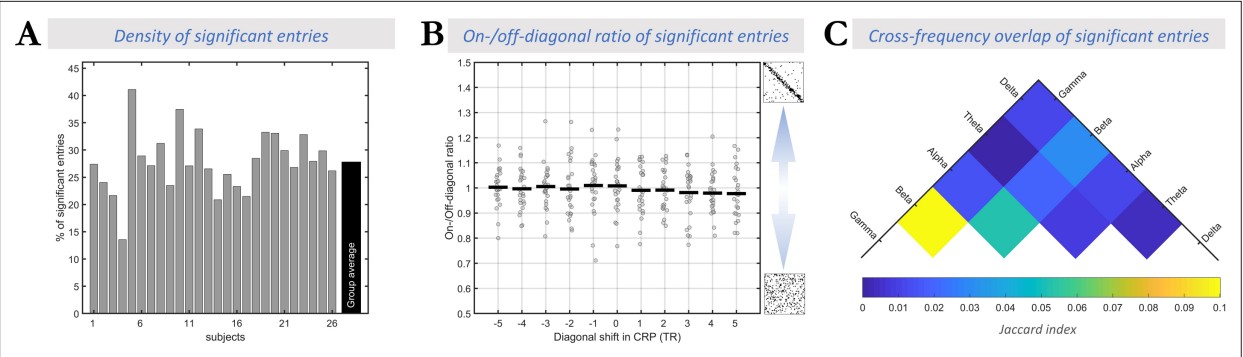

**Figure 4.** Findings generalize to the whole-brain connectome in concurrently recorded source-localized scalp EEG-fMRI. (**A**) Total significance rate of the cross-modal spatial similarity passing significance testing in the multi-frequency cross-modal recurrence plot (CRP), shown across all subjects (N=26; analogous to *Figure 3B*). (**B**) The on-/off-diagonal ratio of the rate of spatially correlated epochs in the multi-frequency CRP for different diagonal shifts (analogous to *Figure 3C*). Note that lag of zero corresponds to the original preprocessed data where fMRI has been shifted 6 s to account for the hemodynamic delay. (**C**) Overlap of spatially correlated epochs across pairs of electrophysiological frequencies in the multi-frequency CRP (averaged over subjects; analogous to *Figure 3D*). Equivalent findings are presented for EEG phase coupling connectomes as a supplemental analysis (see Appendix 1 and *Figure 3—figure supplement 4*).

HRF-convolved iEEG revealed on-/off-diagonal ratios close to 1 (mean real ratio = 1.02 ± 0.12 vs. mean null ratio = 1.00 ± 0.02), and a $BF_{01}$=2.0 indicated that the data are twice as likely to occur under H0 (aligned with scenario II) than under H1 (aligned with scenario I). These observations were replicated across a wide range of diagonal shifts in CRP from –15s to 15s equivalent to 5TRs ($BF_{01}$=3.2 ± 1.95; ratio averaged over shifts = 1.02 ± 0.04, null ratio averaged over shifts = 1.02 ± 0.01). The lack of temporal convergence using HRF-convolved iEEG implies that fMRI connectome dynamics are not merely a temporally restricted or smoothed version of electrophysiological dynamics.

The observed temporal divergence of large-scale connectome organization even after HRF convolution of iEEG signals may be surprising given that prior literature supports the existence of an electrophysiological basis for fMRI FC dynamics (*Wirsich et al., 2020*; *Zhang et al., 2020*). In line with this prior work, we replicated the known *connection-level* temporal convergence in our intracranial EEG-fMRI data (section III of Appendix 1). Specifically, we found significant cross-modal temporal association in a substantial proportion (albeit not all) individual connections over all frequency bands (*Figure 3—figure supplement 2*). Our findings confirm the existence of electrophysiological correlates of fMRI FC dynamics at the level of *individual connections*, while suggesting the presence of additional neural processes that drive independent trajectories of connectome-level FC *patterns* across the two modalities.

## Generalization to whole-brain connectomes in source-localized EEG-fMRI

To validate our results in the whole-brain connectome, we analyzed a resting state concurrent source-localized EEG-fMRI dataset composed of 26 neurologically healthy subjects (*Figure 4*). Replication in source-orthogonalized and leakage-corrected data is provided in *Figure 3—figure supplement 3*.

The asynchronous analysis and ensuing CRPs confirmed our observations of the intracranial dataset. As shown in *Figure 4A*, 28 ± 6% of CRP entries showed statistically significant correlation values. Again, the group-average on-/off-diagonal ratio was close to 1 (*Figure 4B*), favoring the view of asynchronous convergence across fMRI and EEG connectome patterns. In particular, a group-level one-tailed paired *t*-test of the observed ratios against the null ratios (average across random permutations of each subject) did not show any significant differences (1.01±0.1 vs. null 1.00±0.01; $t_{25}$=0.43; *P*=0.34). Bayesian negative one-tailed paired *t*-test with an average of $BF_{01}$=3.4 provided evidence that scenario II (H0) is three times more likely than scenario I (H1). This result held irrespective of assumed delay between fMRI and EEG over a wide range of diagonal shifts in CRP (excluding zero lag) including from –10s to 10s equivalent to 5TRs ($t_{25}$=−1.00 ± 0.76; FDR-corrected p>0.98; $BF_{01}$=8.9 ± 3.7; ratio averaged over shifts = 1.00 ± 0.01, null ratio averaged over shifts = 1.01 ± 0.00). This observation emphasizes *temporal* divergence of the FC processes captured by the two modalities.

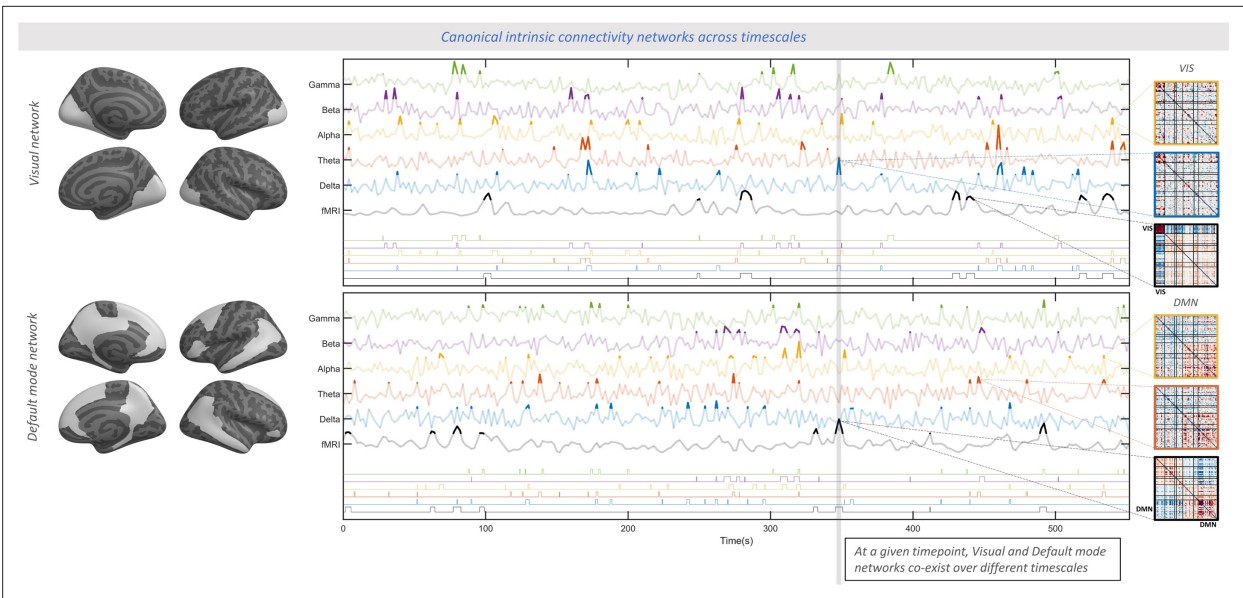

**Figure 5.** Emergence of canonical intrinsic connectivity networks (ICNs) across the breadth of functional connectivity timescales. In a sample subject, overall connectivity strength within the Visual network (top) and the Default mode network (bottom) are separately shown for all functional connectivity (FC) timescales during the whole recording period (fMRI-FC and EEG-FC$_{Amp}$, averaged across all region pairs within the respective network). Left: canonical Visual and Default mode networks (*Yeo et al., 2011*) rendered on a cortical surface. Center: The timeseries show the continuous changes of FC strength (z-scored for better visualization) for all timescales sorted from slow to fast in ascending order, using the same color coding as in *Figure 2*. Thresholded strength of each network (z>2) is shown for all timescale in a similar fashion on the bottom section of the plot to emphasize periods of particularly high connectivity. The gray vertical bar marks a sample timepoint where the two networks emerge concurrently at two different timescales: the Visual network in the EEG-theta band and the Default mode network in fMRI. This observation demonstrates the co-occurrence of multiple distinct networks at different FC timescales at one given timepoint. Right: The matrices show the whole-brain spatial FC patterns averaged over the thresholded timepoints (z>2) of the respective timescale. The cross-timescale spatial similarity is visually apparent among the top three (respectively bottom three) matrices. The complete set of FC timescales is shown in *Figure 5—figure supplement 1*. In line with scenario II, these spatially similar FC patterns (dominated by high FC within the Visual or Default mode network, respectively) occur asynchronously across timescales. Please note that this figure aims at making the cross-timescale spatial similarity of connectomes visually accessible and does not reflect a quantitative analysis. For the statistical approach and analyses of this study, see *Figures 2–4*.

The online version of this article includes the following figure supplement(s) for figure 5:

**Figure supplement 1.** The matrices are the same as shown on the right side of *Figure 5* in the main text, but for all timescales (fMRI and δ-γ in EEG).

As in the intracranial dataset, the Jaccard index for the overlap of significant epochs across frequency-specific CRPs was low (maximum value: 0.10±0.03 for β- vs. γ-band CRPs; *Figure 4C*), speaking to the multi-frequency nature of the association between fMRI and EEG connectome dynamics. In summary, these findings are in line with our observations in the intracranial EEG-fMRI dataset.

We illustrate the multi-frequency and asynchronous nature of the observed cross-modal convergence in a sample subject (*Figure 5*). To this end, we extracted the within-network connectivity strength of two canonical intrinsic connectivity networks (ICNs), prominent features of the connectome's architecture (*Yeo et al., 2011*). While individual ICNs each capture only part of the connectome, their well-known role in cognition helps understand the functional implications of our observations. The data shows that each ICN emerges at different timepoints across FC timescales, illustrating the observed multi-timescale nature of connectome reconfigurations. Such asynchronicity independently enables FC within multiple ICNs at any given time, each at a different FC timescale (vertical gray bar in *Figure 5*).

## Cross-modal convergence arises from discrete recurrent connectome states

It is well-known that static (time-averaged) connectomes are correlated across fMRI and iEEG (*Betzel et al., 2019*) as well as scalp EEG (*Wirsich et al., 2021*). Therefore, the observed cross-modal convergence in the CRPs could simply reflect the varying degree to which the cross-modally shared static

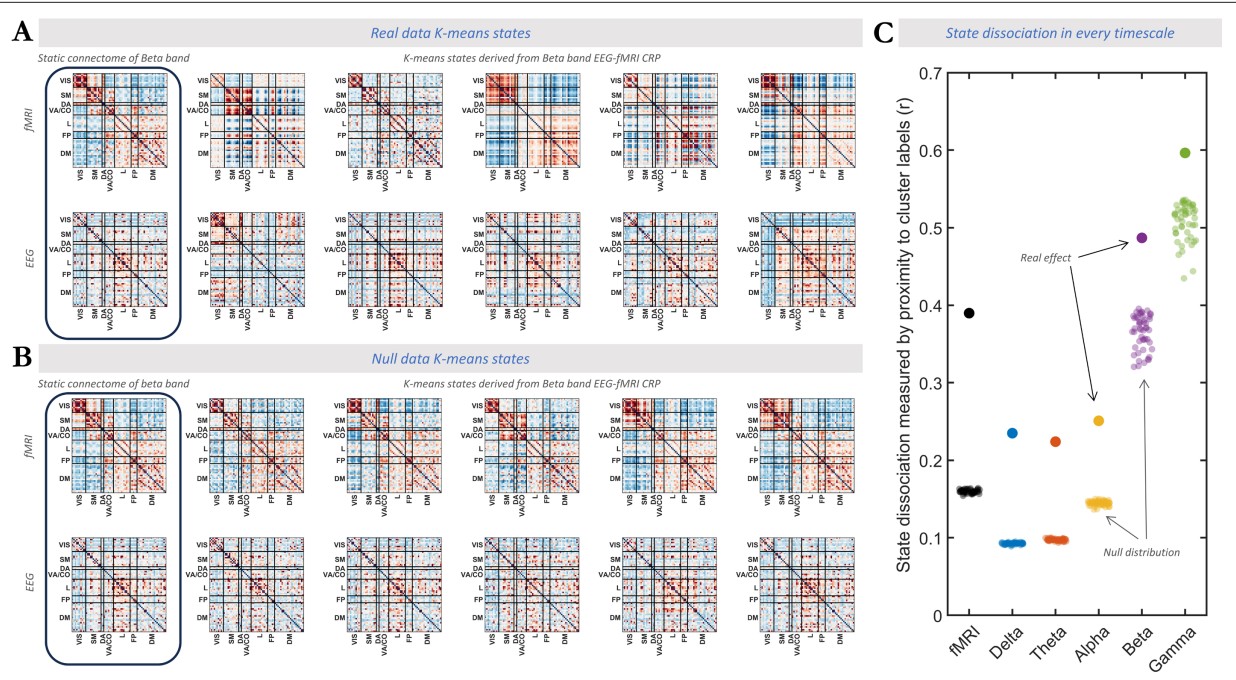

**Figure 6.** fMRI and EEG connectome convergence arises from distinct recurrent states. (**A**) Left: Group-level static connectomes of fMRI (top) and EEG (bottom) are illustrated. Right: Five dissociable connectome states at the group level were identified for fMRI and EEG (depicted for β band as an example). Red and blue colors represent positive and negative connectivity values, respectively. (**B**) Similar to A but for a sample surrogate dataset with randomly phase-permuted connectivity timecourses. Note that states in the surrogate data are highly similar to one another and the static connectome. (**C**) The vertical axis shows the extent to which the detected connectome states within a functional connectivity (FC) timescale are dissociable. The state dissociation was calculated by the similarity between the proximity matrix of the empirical data and the proximity matrix extracted from cluster labels (a binary proximity matrix representing complete dissociation). The horizontal axis shows timescales (fMRI and EEG frequency bands) on which the clustering was performed. Filled circles show the real effect while the transparent circles show surrogate data. In every timescale, K-means states in the real data were statistically more dissociable than every single surrogate sample (z=103.6, 128.0, 91.2, 35.5, 5.5, 3.7 for fMRI and δ- to γ-bands of EEG, respectively). These observations speak to the presence of distinct recurrent patterns among the converging connectome patterns across timescales.

The online version of this article includes the following figure supplement(s) for figure 6:

**Figure supplement 1.** Replication of clustering analysis with number of states equal to seven.

connectome organization is expressed at any timepoint. As a first step to rule out this possibility, we demonstrated that epochs of high spatial similarity across data modalities do *not* temporally align with moments of particularly strong expression of the static connectome organization in either data modality (Appendix 1 and *Figure 3—figure supplement 1*).

Next, we show that the FC patterns in each data modality reflect recurrent connectome states that differ from each other and the static connectome. States were identified in a data-driven manner separately in fMRI and scalp EEG (group-level K-means clustering of frame-wise connectome patterns with K=5 and K=7 for replication; *Figure 6A*). Proximity matrices quantified within- and between-cluster similarity of samples. We repeated the process for surrogate FC data generated by a temporal phase permutation approach that preserves static connectome organization while destroyed the dynamic connectome patterns (*Allen et al., 2014*; a sample shown in *Figure 6B*). As shown in *Figure 6C*, the dissociability between states fell above the null distribution for fMRI (z=103.6) and EEG (z=128.0, 91.2, 35.5, 5.5, 3.7 for δ- to γ-bands). Similar results were obtained with seven states (*Figure 6—figure supplement 1*). These findings extend prior observations of recurrent states in fMRI and source-space EEG connectomes (*Allen et al., 2014*; *Abreu et al., 2021*) in support of their veridic spatial convergence across modalities.

## Addressing alternative sources of contribution

We aimed at ruling out that fluctuations in signal-to-noise ratio (SNR) of FC estimates, head motion, epileptiform activity (for intracranial data), or volume conduction (for scalp data) drive the observed

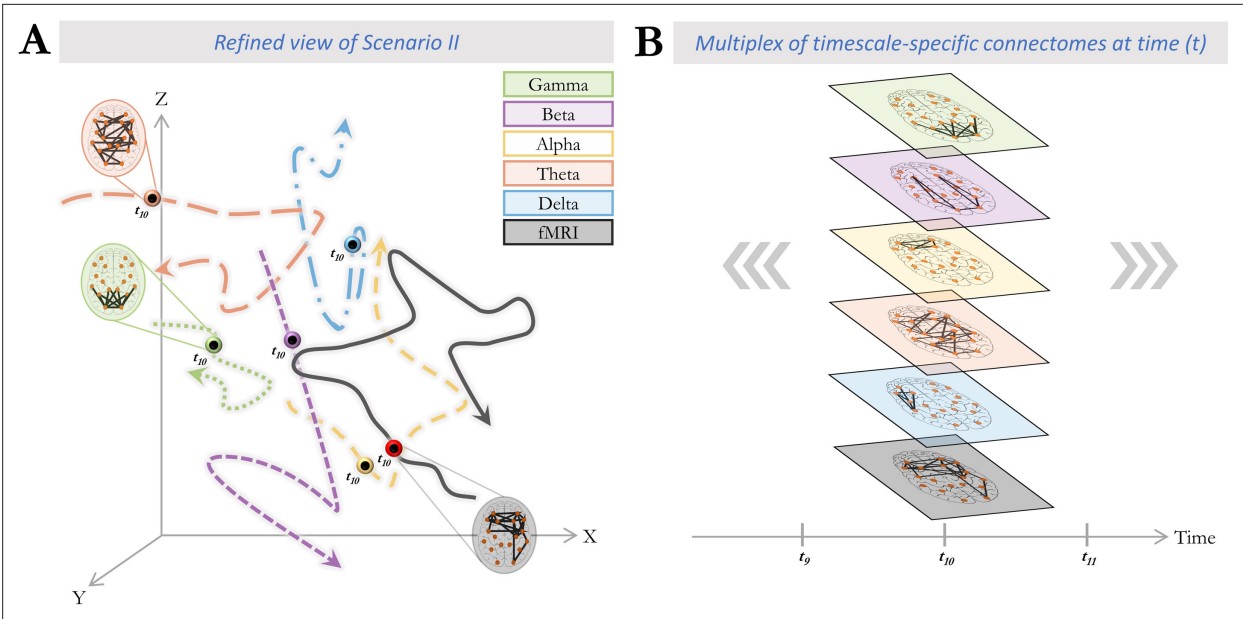

**Figure 7.** Multiplex of dynamic connectome trajectories across timescales of functional connectivity captured by fMRI and electrophysiology. Our findings indicate that connectome dynamics across fMRI and each frequency band of electrophysiology are spatially convergent (comprise sets of similar spatial states) but temporally divergent (the states occur at different times). Co-existence of such temporally divergent connectome trajectories across functional connectivity (FC) timescales is consistent with a multiplex of connectome patterns at any particular time point. (**A**) Each color-coded trajectory represents the dynamic spatial reconfigurations of connectome patterns of a particular timescale in state space, within a hypothetical time period. The dots marked on each trajectory represent the same instant in time (e.g. $t_{10}$), highlighting the different position that each trajectory occupies in state space at this instant. Note that trajectories are partially occupying the same area of the state space (c.f. scenario II in *Figure 1*). (**B**) An example of the co-existence of connectome patterns across different timescales embedded within a multiplex network at the exemplary timepoint $t_{10}$. Functionally, this multiplex of connectome configurations may serve as parallel 'communication channels' that maximize information flow between brain regions, akin to the multitude of independent information streams multiplexed into a single radio or television transmission medium.

The online version of this article includes the following figure supplement(s) for figure 7:

**Figure supplement 1.** Alternative sources of contributions to the cross-modal similarity timecourse.

association between the two modalities or the sparsity thereof. To this end, we tested whether the timecourse of each of these factors aligns with the timecourse of cross-modal spatial similarity (Appendix 1 and *Figure 7—figure supplement 1*). We found no such temporal relationship in either intracranial or scalp EEG-fMRI datasets. The temporal independence of cross-modal similarity from these artifacts speaks to a neural basis of the observed spatial convergence and temporal divergence of connectome reconfigurations across timescales.

## Discussion

We set out to assess whether connectome dynamics comprise a *multiplex* of parallel network trajectories reflecting different FC timescales. We capitalized on rare concurrent intracranial EEG and fMRI in humans with replication in source-localized scalp EEG-fMRI. Our core finding was that fMRI and each canonical EEG frequency band capture spatially similar recurrent connectome states that, however, occur asynchronously. This finding is in line with scenario II described in *Figure 1*, but with the refinement that state trajectories are largely temporally independent not only across data modalities (hemodynamic vs. iEEG/EEG) but also across band-specific electrophysiological processes. The concurrent presence of temporally independent trajectories (*Figure 7A*) implies the co-expression of multiple connectome configurations at any given timepoint (exemplified as $t_{10}$ in *Figure 7B*).

Our findings render it unlikely that hemodynamics- and electrophysiology-derived connectivity reconfigurations reflect a single underlying neural process. In particular, if fMRI and EEG merely provided two different windows onto the same underlying processes, one would expect these processes to appear in the two data modalities around the same time (albeit with a possible delay). Instead, the observed *asynchronicity* of spatially similar connectome patterns is in line with the

viewpoint that the data modalities are more sensitive to different aspects of neural processing. This view aligns with the standpoint that electrophysiology better captures neural activity involving fast-conducting (i.e. thick, myelinated) fibers, while fMRI prominently reflects neural ensembles connected via slow (i.e. thin, unmyelinated) fibers (*Hari and Parkkonen, 2015*). According to this standpoint, each type of neural process, slow and fast, subserves cognitive processes at different speeds, suggesting that each modality better captures particular aspects of behavioral neural correlates (*Hari and Parkkonen, 2015*). Extending this view to connectomics, distinct connectome patterns reflecting partly non-overlapping FC processes could dominate the signals in hemodynamic and electrophysiological acquisition methods, respectively.

This perspective could also explain why spatial similarity across hemodynamics and electrophysiological connectomes rarely surpass a small effect size. These effect sizes are consistently small for cross-modal comparisons in both static (*Betzel et al., 2019*; *Wirsich et al., 2021*) as well as time-varying connectomes (*Wirsich et al., 2020*). In line with these prior reports, we acknowledge that the cross-modal spatial similarity of connectome frames is small albeit above chance in the current study. The small effect size implies that beyond the common spatial patterns, connectome-wide co-activation patterns contain a substantial amount of variance that is unique to each data modality and FC timescale (i.e. part of the variance aligns with scenario III). From the above-described standpoint of partly distinct neural substrates, the diverging variance in EEG and fMRI is at least partly neural in nature.

Prior multi-modal studies of neural dynamics have predominantly aimed at methodologically cross-validating hemodynamic and electrophysiological observations, thus focusing on their convergence. These important foundational studies include e.g., the cross-modal comparison of region-wise (*Mukamel et al., 2005*; *Nir et al., 2007*) or ICN-wise (*Mantini et al., 2007*) activity fluctuations, instantaneous activity maps (*Hunyadi et al., 2019*; *Zhang et al., 2020*) or EEG microstates (*Van de Ville et al., 2010*), infraslow connectome states (*Abreu et al., 2020*), or connection-wise FC including studies in the iEEG-fMRI and scalp EEG-fMRI data used in the current study (*Ridley et al., 2017*; *Wirsich et al., 2020*, respectively). In contrast to this prior work, the current study investigated the highly time-resolved cross-modal temporal relationship at the level of FC *patterns* distributed over all available pairwise connections, and found a *connectome-level* temporal divergence. The discrepancy between temporal divergence in our study and convergence in prior studies implies that infraslow fluctuations of activity in *individual* regions or of FC in individual region-pairs observable in both modalities (prior studies) are neurally distinct from *connectome-wide* FC dynamics observable separately in each modality (current study). Indeed, we confirmed the existence of infraslow electrophysiological FC dynamics driving cross-modal temporal associations at the level of individual connections (*Figure 3—figure supplement 2*). Of note, this significant temporal association occurred across many but not all connections, leaving room for dissociation at the level of large-scale FC patterns. Generative models may shed light on the factors underlying the dissociation of connectivity processes at the level of connectome patterns versus individual connections (*Deco et al., 2009*; *Cabral et al., 2014*; *Rabuffo et al., 2021*).

Another core conclusion pertains to the rich multi-frequency signature of the cross-modal relationship. Specifically, in both synchronous and asynchronous approaches, hemodynamic FC configurations spatially converged with electrophysiological FC patterns in all canonical frequency bands and did so at largely non-overlapping timepoints across bands. Similarly, in our clustering approach, all four fMRI connectome states matched with specific EEG connectome states in all frequency bands, rather than each individual fMRI state being tied to a specific EEG frequency band. This observation is in line with our independent, unimodal iEEG study, showing that the connection-wise dynamics of electrophysiological FC are temporally independent across frequency bands (*Mostame and Sadaghiani, 2021*). These observations suggest that electrophysiological connectome dynamics do not constitute a unitary FC process, but rather a set of frequency-specific FC processes unfolding in a dissociable and complementary manner. Note that although our approach was motivated by frameworks of oscillation-based FC mechanisms (*Engel et al., 2013*), our conclusion of frequency-specificity does not hinge upon exclusively oscillatory processes and is also compatible with aperiodic fluctuations (*Donoghue et al., 2020*).

Overall, the co-existence of multiple distinct connectivity processes across timescales (from fMRI to faster electrophysiological frequency bands) may be conceptualized as a multiplex system. A

multiplex architecture allows the system nodes (here brain regions) to connect to one another over multiple layers, where each layer represents a type (here timescale) of connectivity. In this conceptualization, the brain leverages the co-existing timescale-specific communication channels to yield a maximized bandwidth for information flow at any given time. As an example, *Figure 5* illustrates the overall connectivity strength of the Default Mode (DM) and Visual (Vis) ICNs over time and across timescales, for a single subject. Note that scalp EEG-fMRI data was used for this purpose given its full-cortex coverage. It is apparent that each of these networks dominates the FC pattern at different timepoints across timescales, supporting the above-described multiplex nature of the brain connectome dynamics.

## Limitations

The hemodynamic response of the brain bounded the temporal resolution of the results. However, this methodological limitation of fMRI may produce high sensitivity to particularly slow brain processes (*Hari and Parkkonen, 2015*). Similarly, while the temporal resolution of iEEG connectivity was high, it was subject to the temporal smoothing effect of the estimation window. Oscillation-based connectivity requires several oscillation cycles to yield a reliable connectivity estimate thus necessitating an estimation window of several seconds in continuous recordings. While the EEG estimation window may be considered a methodological limitation, it aligns with the neurobiological viewpoint that several oscillation cycles are needed for cross-region synchronization to be functionally effective.

Another consideration is that we could not reliably investigate high-frequency broadband ('high-γ' band; >60 Hz) range, because concurrently recorded iEEG is disproportionately affected by MR-related artifacts in higher frequencies (*Mullinger and Bowtell, 2011*; *Mele et al., 2019*). Static FC based on high-frequency broadband amplitude fluctuations may match fMRI-derived FC particularly well (*Kucyi et al., 2018*) and could potentially result in connectome dynamics temporally convergent with those in fMRI (if their concurrent measurements were feasible). Importantly, however, this possibility is not inconsistent with our observations that electrophysiological FC patterns in delta through gamma bands constitute parallel dynamics that are temporally independent of each other and of fMRI.

The limited spatial sampling (coverage) of iEEG electrodes in patients with epilepsy - dictated by clinical considerations- is another important limitation. Nonetheless, we were able to replicate our findings in the absence of such limited spatial sampling by leveraging whole-brain source-localized EEG-fMRI recordings. In iEEG data, volume conduction effects are minimal (*Rouse et al., 2016*; *Dubey and Ray, 2019*) and source leakage correction was performed in the source-localized EEG recordings (*Colclough et al., 2015*). It should be noted, however, that even state-of-the-art volume conduction mitigation strategies cannot distinguish artefactual zero-phase lag from genuine zero-phase lag neural activity. To mitigate this issue, we provide results from source-localized data both with and without leakage correction (*Figure 3—figure supplement 3* and main text, respectively).

## Conclusions

Based on our multi-modal observations, we reconceptualize the functional connectome as a composite process that engages a shared repertoire of recurrent configurations across a maximally broad range of timescales (*Figure 5A*). This process is akin to frequency multiplexing in radio and television broadcasting where multiple streams of information are transmitted simultaneously over one communication system (the telecommunication network), taking advantage of distinct bands across the frequency domain. In this conceptualization, information flow is maximized as the connectome can concurrently take advantage of several connectivity states at any given moment (*Figure 5B*). Such multiplexed FC architecture is thus optimally poised to support the multiple speeds of complex brain function. To conclude, our findings conceptually advance the understanding of the functional capabilities that large-scale connectivity affords. This advance motivates the use of multimodal approaches in future connectomics research to accommodate the multi-timescale nature of FC.

## Methods

This study quantifies the spatial and temporal convergence of dynamic reconfigurations of connectomes driven by different FC timescales (*Figure 1*). Note that here timescale refers to FC rather than its dynamics. More specifically, timescale denotes the speed of the neuronal activity from which FC is

derived, i.e., particular electrophysiological frequency bands or the infraslow scale of hemodynamic fluctuations. It does not refer to the speed at which FC patterns reconfigure. We assess whether fMRI and electrophysiology connectome reconfigurations are: (I) spatially and temporally convergent, (II) temporally divergent but spatially convergent, and (III) spatially and temporally divergent. We did not include the scenario where temporal convergence exists without spatial convergence. Assessing this question would require a distinct set of exploratory analyses covering a very large set of features in each data modality, and is therefore outside the scope of this study.

This study utilized two independent datasets: (1) concurrent fMRI and intracranial EEG resting state recordings of nine subjects with drug-resistant epilepsy, and (2) concurrent fMRI and source-localized EEG resting state recordings of 26 healthy subjects. Findings established in the first dataset under conditions of minimal volume conduction were subsequently generalized to whole-brain connectomes. Note these datasets comprise different (clinical and non-clinical) populations and were acquired with different hardware and imaging sequences.

## Data and subjects
### Intracranial EEG-fMRI dataset
The data has been originally introduced elsewhere (*Ridley et al., 2017*). Briefly, nine patients (average 30.4±4.5 y; range 24–38; three females) undergoing presurgical monitoring for treatment of intractable epilepsy gave informed consent according to procedures approved by the Joint Research Ethics Committee of the National Hospital for Neurology and Neurosurgery (NHNN, UCLH NHS Foundation Trust) and UCL Institute of Neurology, London, UK (*Carmichael et al., 2010*; *Carmichael et al., 2012*).

Data acquisition is detailed in *Ridley et al., 2017*. Briefly, fMRI data was acquired using a 1.5T Siemens Avanto scanner with a GE-EPI pulse sequence (TR = 3 s; TE = 78 ms; 38 slices; 200 vol; field of view: 192×192; voxel size: 3×3×3 mm$^3$). Structural T1-weighted scans were acquired using a FLASH pulse sequence (TR = 3 s; TE = 40 ms; 176 slices; field of view: 208×256; voxel size: 1×1×1.2 mm$^3$). Intracranial EEG was recorded using an MR-compatible amplifier (BrainAmp MR, Brain Products, Munich, Germany) at 5 kHz sampling rate. Number of electrodes (ECoG (grid/strip) and depth electrodes) and the associated fMRI regions of interest (ROIs) that were included in our analysis was on average ~37 (Min = 11; Max = 77; Median = 36). *Figure 1—figure supplement 1* shows the location of the implanted depth and ECoG electrodes for each subject. The data was acquired during a 10 min resting state scan (except for subject P05: ~5 min). Subjects were instructed to keep their eyes open.

### Scalp EEG-fMRI dataset
The data has been originally introduced elsewhere (*Sadaghiani et al., 2010*). Briefly, 10 min of eyes-closed resting state were recorded in 26 healthy subjects (average age = 24.39 y; range: 18–31 y; eight females) with no history of psychiatric or neurological disorders. Informed consent was given by each participant and the study was approved by the local Research Ethics Committee (CPP Ile de France III). FMRI was acquired using a 3T Siemens Tim Trio scanner with a GE-EPI pulse sequence (TR = 2 s; TE = 50 ms; 40 slices; 300 vol; field of view: 192×192; voxel size: 3×3×3 mm$^3$). Structural T1-weighted scans were acquired using the MPRAGE pulse sequence (176 slices; field of view: 256×256; voxel size: 1×1×1 mm$^3$). 62-channel scalp EEG (Easycap, with an additional EOG and an ECG channel) was recorded using an MR-compatible amplifier (BrainAmp MR, Brain Products) at 5 Hz sampling rate.

## Data preprocessing
### Intracranial EEG-fMRI dataset
The fMRI data was preprocessed as explained in *Ridley et al., 2017*. Briefly, for each subject, common fMRI preprocessing steps were performed in SPM8 (https://www.fil.ion.ucl.ac.uk/spm/software/spm8/) (Respectively: slice time correction, realignment, spatial normalization, and smoothing of 8 mm). Then, the BOLD signal timecourse at each voxel was detrended and filtered (0.01–0.08 Hz). Using Marsbar toolbox in SPM (http://marsbar.sourceforge.net/), spurious signal variations were removed by regressing out the signals of lateral ventricles and the deep cerebral white matter. After preprocessing, functional data was co-registered to the structural T1-weighted scan of the subject. Then at the location of each iEEG electrode (see below), the BOLD signals of adjacent voxels within a 5 mm radius was averaged to generate the region of interest (ROI) BOLD signal corresponding to that iEEG

electrode (Similar to *Kucyi et al., 2018*). To minimize spatial overlap of fMRI ROIs, adjacent ROIs with less than 9 mm distance were excluded from the functional connectomes of both iEEG and fMRI.

T1-space position of iEEG electrodes of each subject were estimated once the post-implantation CT scan of each subject was co-registered to their structural T1-weighted scan (*Ridley et al., 2017*). Then, the MNI location of the electrodes in each subject was calculated (shown in *Figure 1—figure supplement 1*). Intracranial EEG data was corrected for gradient and cardio-ballistic artifacts using Brain Vision Analyzer software (*Allen et al., 2000*) and down-sampled to 250 Hz. Proceeding with a spike detection analysis (*Bettus et al., 2011*; *Ridley et al., 2017*), electrodes that were marked by clinicians as involved in generating seizures or generating interictal spikes were excluded from further analyses. To remove slow drifts, line noise, and high-frequency noise, the data was filtered with a fourth-order high-pass Butterworth filter at 0.5 Hz, a sixth-order notch filter at 50 Hz, and a fourth-order low-pass Butterworth filter at 90 Hz, respectively. Since the interpretation of white matter BOLD signals is debated (*Gawryluk et al., 2014*), depth electrodes that were implanted in the white matter (and their corresponding fMRI ROIs) were excluded from the analyses. For this purpose, we excluded electrodes that were embedded outside of SPM's gray matter template mask in MNI space. Additionally, electrodes with remaining jump artifacts, highly prevalent periods of epileptiform activity, or low SNR were removed by visual inspections under supervision of an epileptologist. After this cleaning procedure, electrodes were re-referenced to the common average. Note that in order to preserve temporal continuity of the concurrent recordings, we did not exclude time intervals of the iEEG data that contained additional interictal activity in the healthy brain regions. However, to ensure that interictal activity had no critical effect on our findings, we generalized our observations in a non-clinical population (source-localized EEG-fMRI dataset) and further statistically demonstrated in the intracranial data the temporal independence of cross-modal similarity of connectome dynamics from visually marked epochs of interictal activity (19 ± 15% of data length) (see section 'Addressing alternative sources of contribution' in the Appendix 1).

## Scalp EEG-fMRI dataset

fMRI and EEG data were preprocessed with standard preprocessing steps as explained in detail elsewhere (*Wirsich et al., 2020*). In brief, fMRI underwent standard slice-time correction, spatial realignment (SPM12, http://www.fil.ion.ucl.ac.uk/spm/software/spm12). Structural T1-weighted images were processed using Freesurfer (recon-all, v6.0.0, https://surfer.nmr.mgh.harvard.edu/) in order to perform non-uniformity and intensity correction, skull stripping, and gray/white matter segmentation. The cortex was parcellated into 68 regions of the Desikan-Kiliany atlas (*Desikan et al., 2006*). This atlas was chosen because —as an anatomical parcellation— avoids biases towards one or the other functional data modality. The T1 images of each subject and the Desikan-Killiany were co-registered to the fMRI images (FSL-FLIRT 6.0.2, https://fsl.fmrib.ox.ac.uk/fsl/fslwiki). We extracted signals of no interest such as the average signals of cerebrospinal fluid (CSF) and white matter from manually defined regions of interest (ROI, 5 mm sphere, Marsbar Toolbox 0.44, http://marsbar.sourceforge.net) and regressed out of the BOLD timeseries along with 6 rotation, translation motion parameters, and global gray matter signal (*Wirsich et al., 2017*). Then we bandpass-filtered the timeseries at 0.009–0.08 Hz. Average timeseries of each region was then used to calculate connectivity.

EEG underwent gradient and cardio-ballistic artifact removal using Brain Vision Analyzer software (*Allen et al., 1998*; *Allen et al., 2000*) and was down-sampled to 250 Hz. EEG was projected into source space using the Tikhonov-regularized minimum norm in Brainstorm software (*Baillet et al., 2001*; *Tadel et al., 2011*). Source activity was then averaged to the 68 regions of the Desikan-Killiany atlas. Band-limited EEG signals in each canonical frequency band and every atlas region were then used to calculate frequency-specific connectome dynamics. Note that the MEG-ROI-nets toolbox in the OHBA Software Library (OSL; https://github.com/OHBA-analysis/osl-ephys; *Quinn et al., 2025*) was used to minimize source leakage (*Figure 3—figure supplement 3*) in the band-limited source-localized EEG data (*Colclough et al., 2015*).

## Connectivity analyses

Connectivity analyses were identical in the two datasets unless stated otherwise.

## Dynamic measures of connectivity

Amplitude- and phase coupling (EEG-FC and EEG-FC$_{\text{Phase}}$) are two measures of functional connectivity in neurophysiological data reflecting distinct coupling modes (**Mostame and Sadaghiani, 2020**). In this study, we use both EEG-FC$_{\text{Amp}}$ (for the main results) *and* EEG-FC$_{\text{Phase}}$ (for replication purposes, see **Figure 3—figure supplement 4**) and collectively refer to them as *iEEG FC* (or *EEG FC* for the source-localized scalp dataset). We investigate five canonical electrophysiological frequency bands: δ (1–4 Hz), θ (5–7 Hz), α (8–13 Hz), β (14–30 Hz), γ (31–60 Hz). To estimate iEEG/EEG FC at each canonical frequency band, we first band-passed the electrophysiological signals using a fourth-order Chebyshev type II filter. We used the Hilbert transform of the band-passed signals to extract the envelope and unwrapped phase of each signal.

Amplitude coupling (EEG-FC$_{\text{Amp}}$) was defined as the coupling of the z-scored activation amplitude of the two distinct electrodes, within the time window of FC estimation. Specifically, EEG-FC$_{\text{amp}}$ between electrodes *i* and *j* at time *t* over the frequency band *freq* was calculated as:

$$EEG\_FC_{Amp_{ij}^{freq}}(t) = \frac{1}{N} \sum_{m=t-\frac{L}{2}}^{t+\frac{L}{2}} Z\left(env_i^{freq}\right)(m) \times Z\left(env_j^{freq}\right)(m)$$

where *L* is the window length equal to TR (3 s for the intracranial data and 2 s for the scalp data), *N* is the number of data points within the window, and *Z(env)* is the envelope of the signal that is Z-scored with respect to its timepoints across the whole time of data acquisition. Note that our approach for estimating EEG-FC$_{\text{Amp}}$ is a modified version of the fine-grained fMRI-FC measure recently introduced by **Zamani Esfahlani et al., 2020**. Given that iEEG signals are considerably faster than BOLD signals by nature, here the measure is averaged over *N* consecutive samples within the window to increase SNR. The center of the window (timepoint *t)* stepped at every TR, providing an exact temporal match to original fMRI time steps.

Major findings were replicated using an alternative mode of electrophysiology-derived connectivity, phase coupling (see **Figure 3—figure supplement 4**). EEG-FC$_{\text{Phase}}$ was defined as the consistency of the phase difference of band-passed signals across an electrode pair, within the time window of length *L* equal to TR:

$$EEG\_FC_{phase_{ij}^{freq}}(t) = \frac{1}{N} \sum_{m=t-\frac{L}{2}}^{t+\frac{L}{2}} e^{j\Delta\varphi_{ij}^{freq}(m)}$$

where $\Delta\varphi$ is the phase difference between the two signals.

The dynamic FC in fMRI data (fMRI-FC) was estimated using the original fine-grained measure of FC by **Zamani Esfahlani et al., 2020**. *fMRI-FC* of ROIs *i* and *j* at time *t* was estimated as below, where *Z*(BOLD) is the Z-scored BOLD signal of the ROI with respect to its own timepoints.

$$fMRI\_FC_{ij}(t) = Z\left(BOLD_i(t)\right) \times Z\left(BOLD_j(t)\right)$$

Finally, to compensate for the time lag between hemodynamic and neural responses of the brain (**Logothetis et al., 2001**), we shifted the fMRI-FC timecourse 6 s backwards in time.

In fMRI connectome computation, some prior work has used partial correlation instead of full correlation. Partial correlation emphasizes direct connections by calculating correlation between any pair of brain regions after regressing out the timeseries of all other regions. However, we have opted to use full correlation because this permits interpretation of our outcomes in the context of the vast existing literature that uses full correlations in fMRI, including the majority of bimodal (EEG-fMRI) connectome studies (e.g. **Tagliazucchi et al., 2012**; **Deligianni et al., 2014**; **Wirsich et al., 2017**; **Wirsich et al., 2020**; **Wirsich et al., 2021**; **Allen et al., 2018**).

## Static measures of connectivity

Static FC of each measure was extracted by averaging the connection-wise FC values across all time-points of dynamic FC for EEG-FC$_{Amp,}$ EEG-FC$_{Phase,}$ and fMRI-FC, respectively.

## Statistical evaluation of CRP epochs

For each subject, we statistically assessed the significance of the cross-modal spatial similarity of connectome configurations at each CRP epoch separately. To this end, we generated a set of randomized FC matrices by spatially phase-permuting the original fMRI FC matrices – at the corresponding timepoint- in 2D Fourier space and then reconstructing the matrices using the inverse 2D Fourier transform (*Prichard and Theiler, 1994*; *Tewarie et al., 2016*; *Wirsich et al., 2017*). A set of surrogate cross-modal spatial correlation values were estimated at each CRP epoch using the described null data. Finally, the original spatial correlation across fMRI and iEEG connectomes was compared to the estimated null distribution of spatial correlation values (100 randomizations). Positive correlation values exceeding the fifth percentile of the null distribution entered multiple comparisons correction according to the number of CRP epochs (i.e. size of CRP = $S \times S$ where $S$ is the number of fMRI volumes during the whole recording), using the Benjamini Hochberg FDR correction method ($q < 0.05$). We did not include negative correlation values in further analyses given that the interpretation of spatial anti-correlation of large-scale connectome organizations may be challenging and beyond the scope of this study.

The above-described approach resulted in one CRP matrix per subject, EEG frequency band, and electrophysiological connectivity mode (EEG-FC$_{Amp}$ and EEG-FC$_{Phase}$). The frequency band-specific CRPs for each connectivity mode were overlaid into multi-frequency CRPs. None of the statistical tests on the CRP depend on sampling density. At the level of individual subjects, the ratio between the significance rate of on- and off-diagonal CRP epochs (on-/off-diagonal ratio) entered a non-parametric test against a null distribution of chance-level on-/off-diagonal ratios. The null distribution was extracted from spatially phase-randomizing the multi-frequency CRPs in the 2D Fourier space and then reconstructing the matrices using the inverse 2D Fourier transform (100 randomizations). We additionally performed a Bayesian test to provide direct statistical evidence in favor of H0 (i.e. cross-modal temporal divergence; on-/off-diagonal ratio ≈ 1) against H1 (i.e. temporal convergence; on-/off-diagonal ratio >1). This statistical test was conducted in JASP software (https://jasp-stats.org/), using the default settings for the prior distribution (Cauchy distribution with scale parameter of 0.707). This analysis was repeated for different diagonal shifts in CRP from −5 to +5 TRs incrementing by 1 TR.

## Assessing inter-band temporal overlap of cross-modal co-occurrences

To find out whether the cross-modal similarities of fMRI and iEEG/EEG connectome dynamics are governed by a multi-frequency link, we employed an inter-band comparison for frequency-specific CRPs. We quantified the extent of inter-band temporal overlap of the significant CRP epochs (colored dots in the multi-frequency CRP in *Figure 3A*) using the Jaccard index:

$$Jaccard\,(A, B) = \frac{\sum \left( A \& B \right)}{\sum \left( A \mid B \right)}$$

Where $A$ and $B$ are 2D CRP matrices with the same size, and & and | are logical 'And' and 'Or,' respectively. Note that the Jaccard index ranges between 0 (for completely non-overlapping vectors) to 1 (for completely matching vectors).

## Co-existence of canonical ICNs in sample data

Upon finding evidence for scenario II, we aimed to visualize in an accessible manner that the connectome architecture leverages a multiplex of patterns across timescales at every timepoint. This was applied to the Visual and Default Mode canonical ICNs of a sample subject in scalp EEG-fMRI data. ICNs were derived by mapping the brain regions (*Desikan et al., 2006*) to a canonical network atlas (*Yeo et al., 2011*). At each timescale, time-varying connectivity strength for an ICN was quantified by averaging the FC among all region pairs of that ICN at each time point, then z-scoring across time-points. The emergence of an ICN was marked as timepoints where its connectivity strength exceeded

an arbitrary value of 2. Note that this approach is for illustration purposes only, complementing the quantitative analyses outlined above.

## Clustering of discrete states and their cross-modal match

This analysis was performed in the source-localized EEG-fMRI dataset only, allowing us to investigate whole-brain connectome states that are shared across subjects. Separately for fMRI and each EEG frequency band, we pooled the connectome configurations of all subjects. Using K-means clustering, we extracted timescale-specific group-level connectome *states*. Given prior work in the literature (*Allen et al., 2014*), we primarily chose five states and replicated our results with seven states. Then within each timescale, we assessed whether the connectome states were distinct from one another.

We used sample-wise proximity matrix to assess K-means performance. A proximity matrix is a matrix with a pairwise distance of data points sorted by cluster labels. This matrix is then compared to a binary matrix where entries corresponding to within-cluster sample pairs are equal to 1 while other entries are equal to 0. Higher correlation between proximity matrix and the ground-truth matrix indicates better dissociation between K-means clustering. This process was repeated for 50 surrogate data generated from randomly phase permuting the connectivity timecourses, which preserves the static connectome but destroys individual connectome configurations. Eventually, the correlation between proximity and ground-truth matrices were compared across real and null data.

## Assessment of potential contribution from artifacts and other sources

To show that the observed cross-modal spatial convergence is not associated with major confounding factors, we quantified in each subject and dataset the temporal correlation between the timecourse of cross-modal spatial similarity at synchronous recording timepoints extracted from the diagonal entries of CRPs, and the timecourses of possible confounding factors including: overall FC strength, head motion, and static connectome 'prominence.' This latter timecourse was calculated as the correlation between frame-by-frame connectome configurations in iEEG or fMRI to the static (i.e. time-averaged) connectome of that modality. The FC strength timecourse was quantified as the root sum squares of the frame-by-frame FC value over all connections of either fMRI or EEG. The head motion timecourse was measured as framewise displacement (FD) (*Power et al., 2012*). We compared the estimated temporal correlation values to corresponding null distributions of correlations generated by phase-permuting the cross-modal spatial similarity timecourse in the Fourier space (see subplots A-C in *Figure 7—figure supplement 1*; group-level *t*-test against subject-specific mean values of 100 surrogate samples). Furthermore, we quantified temporal overlap of the binary timecourse of intervals of visually marked epileptiform activity and the binarized (significance-thresholded) timecourse of cross-modal spatial similarity using the Jaccard index. We compared the outcome to a corresponding null distribution extracted from a temporally-shifted (by a random number of samples between 1 and the length of the original timecourse) version of the binarized cross-modal spatial similarity timecourse (see subplots D in *Figure 7—figure supplement 1*; group-level *t*-test against subject-specific mean values of 100 surrogate samples). Finally, we replicated our major findings in the scalp data using source-orthogonalized signals to show that our findings are largely independent of source leakage (*Figure 3—figure supplement 3*). Particularly, this approach removes any zero-lag temporal correlation between every pair of electrode signals using a multi-variate orthogonalization process so that any spurious correlation due to volume conduction would be minimized (*Colclough et al., 2015*).

## Acknowledgements

This study was supported by a variety of funding sources including Swiss National Science Foundation (SNSF) grants no. 192749 and CRSII5_209470 to SV, the Medical Research Council, United Kingdom, grant G0301067, and the National Institute for Mental Health (1R01MH116226 to Sepideh Sadaghiani)

# Additional information

## Funding

| Funder | Grant reference number | Author |
|---|---|---|
| Swiss National Science Foundation | 192749 | Serge Vulliemoz |
| Medical Research Council, United Kingdom | G0301067 | Louis Lemieux |
| National Institute for Mental Health | 1R01MH116226 | Sepideh Sadaghiani |
| Swiss National Science Foundation | CRSII5_209470 | Serge Vulliemoz |

The funders had no role in study design, data collection and interpretation, or the decision to submit the work for publication.

## Author contributions

Parham Mostame, Conceptualization, Data curation, Software, Formal analysis, Validation, Investigation, Visualization, Methodology, Writing – original draft, Writing – review and editing; Jonathan Wirsich, Conceptualization, Data curation, Software, Formal analysis, Methodology, Writing – review and editing; Thomas Alderson, Ben Ridley, Conceptualization, Data curation, Software, Formal analysis, Investigation, Methodology, Writing – review and editing; Anne-Lise Giraud, Conceptualization, Resources, Data curation, Supervision, Funding acquisition, Investigation, Project administration, Writing – review and editing; David W Carmichael, Serge Vulliemoz, Maxime Guye, Louis Lemieux, Conceptualization, Resources, Data curation, Supervision, Funding acquisition, Investigation, Methodology, Project administration, Writing – review and editing; Sepideh Sadaghiani, Conceptualization, Resources, Data curation, Supervision, Funding acquisition, Validation, Investigation, Visualization, Methodology, Writing – original draft, Project administration, Writing – review and editing

## Author ORCIDs

Parham Mostame ⬛ https://orcid.org/0000-0002-1353-1551
Jonathan Wirsich ⬛ https://orcid.org/0000-0003-0588-9710

## Ethics

Intracranial EEG-fMRI dataset: The data has been originally introduced elsewhere (Ridley et al., 2017). Briefly, nine patients (average 30.4 ± 4.5 years; range 24-38; three females) undergoing presurgical monitoring for treatment of intractable epilepsy gave informed consent according to procedures approved by the Joint Research Ethics Committee of the National Hospital for Neurology and Neurosurgery (NHNN, UCLH NHS Foundation Trust) and UCL Institute of Neurology, London, UK (Carmichael et al., 2010, 2012). Scalp EEG-fMRI dataset: The data has been originally introduced elsewhere (Sadaghiani et al., 2010). Briefly, 10 minutes of eyes-closed resting state were recorded in 26 healthy subjects (average age = 24.39 years; range: 18-31 years; 8 females) with no history of psychiatric or neurological disorders. Informed consent was given by each participant and the study was approved by the local Research Ethics Committee (CPP Ile de France III).

Reviewer #1 (Public review): https://doi.org/10.7554/eLife.98777.4.sa1
Reviewer #2 (Public review): https://doi.org/10.7554/eLife.98777.4.sa2
Author response https://doi.org/10.7554/eLife.98777.4.sa3

# Additional files

## Supplementary files

MDAR checklist

## Data availability

Codes are available online at https://github.com/connectlab/Mostame2024_Multiplex_iEEG_fMRI (copy archived at *Mostame, 2024*). All work in both intracranial EEG-fMRI and scalp EEG-fMRI data constitute secondary analysis of existing data. The details of data availability for each of these data-sets are listed below. The iEEG-fMRI was recorded over an extended period and previously published in several studies (*Vulliemoz et al., 2011*; *Carmichael et al., 2012*; *Ridley et al., 2017*). The participants gave consent for specific research questions as part of an academic research project. Please contact l.lemieux@ucl.ac.uk to inquire about the possibility of collaborations using the iEEG-fMRI data. The scalp EEG-fMRI was previously recorded and published in several papers (*Morillon et al., 2010*; *Sadaghiani et al., 2012*; *Sadaghiani et al., 2010*). Scalp EEG-fMRI data can be openly accessed at the Illinois Data Bank at: https://doi.org/10.13012/B2IDB-6442192_V1.The CRP data of every subject of both datasets, from which most of the findings are driven, are accessible on the Illinois Data Bank server.

The following dataset was generated:

| Author(s) | Year | Dataset title | Dataset URL | Database and Identifier |
|---|---|---|---|---|
| Parham M, Jonathan W, Alderson TH, Ben R, Anne-Lise G, Carmichael DW, Serge V, Maxime G, Louis S, Sepideh S | 2025 | Data for A multiplex of connectome trajectories enables several connectivity patterns in parallel | https://doi.org/10.13012/B2IDB-6442192_V1 | Illinois Data Bank, 10.13012/B2IDB-6442192_V1 |

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

# Appendix 1

## Cross-modal spatial similarity of time-averaged connectome configurations

Prior findings have provided evidence suggesting a robust spatial similarity between the static connectome organization of fMRI and electrophysiology at the group level. In addition to replicating such findings in group-averaged scalp EEG-fMRI connectomes, we replicated these results particularly at the individual level leveraging our unique concurrent intracranial EEG and fMRI data.

In scalp EEG-fMRI data, cross-modal spatial (2D) Pearson correlation of group-level time-averaged connectomes between fMRI and EEG-FC$_{Amp}$ or fMRI and EEG-FC$_{Phase}$ were calculated across all frequency bands. The average spatial correlation value across frequency bands $r=0.28$ and $r=0.28$ for EEG-FC$_{Amp}$ and EEG-FC$_{Phase}$, respectively. The spatial correlation values across all frequency bands and connectivity measures were significantly higher than the corresponding null distributions generated by phase-permuted group-level fMRI-FC spatial organization ($p<0.005$; 200 repetitions; FDR-corrected at $q<0.05$ for the number of frequency bands).

Similarly, in intracranial EEG-fMRI data, cross-modal spatial correlation values of fMRI and iEEG static connectomes of each subject was calculated across frequency bands for both phase- and amplitude-coupling measures (averaged across subjects and frequency bands: $r=0.21$ and $r=0.20$ for EEG-FC$_{Amp}$ and EEG-FC$_{Phase}$, respectively). Subject-specific null distributions in each frequency band and connectivity measure were extracted using phase-permuted FC matrices similar to above. In each frequency band, at the group level, there was a significant effect when comparing subject-specific correlation values with their corresponding average null correlation values. For fMRI-FC to iEEG-FC$_{Amp}$, the t-values of the five frequency bands (δ to γ) were: one-sided $t_8=2.82$, 4.91, 6.19, 5.44, and 6.16 ($p<0.011$; FDR-corrected at $q<0.05$ for the number of frequency bands). For fMRI-FC to iEEG-FC$_{Phase}$, the corresponding values were: one-sided $t_8=2.71$, 3.99, 4.79, 4.36, and 2.67 ($p<0.014$). Altogether, these results suggest that the spatial organization of the intrinsic ('static') architecture in hemodynamics- and electrophysiology-derived connectomes are weakly but significantly correlated. Of note, the small effect sizes are strongly in line with prior literature and may point to possible divergence in the dynamic domain as investigated in the main manuscript.

## Attractors for the cross-modal association

Viewing canonical frequencies from a complimentary perspective reveals another prominent feature of the cross-modal relationship. Specifically, in the multi-frequency CRP, we observed horizontal stripes constructed from non-overlapping frequency-specific epochs of cross-modal spatial convergence (*Figure 2B*). This finding implies that at particular timepoints, the fMRI-derived connectome configuration spatially converges with a component in EEG connectomes that is shared across frequency bands over a large proportion of timepoints. While there is also a degree of vertical stripiness in some multi-frequency CRPs, we quantitatively established the prominence of horizontal CRP stripes in 7/9 of the subjects (*Figure 3—figure supplement 1A*, top). We quantitatively confirmed that the stripes largely correspond to the stable, i.e., static, component of iEEG connectome (*Figure 3—figure supplement 1B–C*). Specifically, we demonstrated the emergence of equivalent stripes in multi-frequency 'pseudo-CRPs' derived from the original fMRI-FC timeseries and a constant timeseries of the time-averaged (static) iEEG connectome. These observations imply that the intrinsic electrophysiological FC component serves as an attractor for the cross-modal spatiotemporal convergence. The considerable number of significant CRP epochs outside the stripes, however, raises the question of whether fMRI and iEEG share spatial patterns distinct from the static connectome attractor, possibly in the form of dissociable recurrent states (see section 'Cross-modal convergence arises from discrete recurrent connectome states' in the main text).

The above-mentioned findings from the dynamic domain provide insight into the static domain. For instance, it has been demonstrated that considering FC matrices of all canonical frequency bands collectively, as opposed to individually, improved prediction of the (group-average) fMRI-derived static connectome. This dovetails with our findings, which suggests that moment-to-moment connectome configurations in individual frequencies each contribute to the static fMRI-derived connectome organization at independent timepoints. In other words, the static fMRI-derived connectome organization comprises the cumulative pattern of all temporally dissociated frequency-specific electrophysiological FC states 'compressed' into a time-averaged architecture.

## Edge-wise association of infraslow FC fluctuations across fMRI and iEEG

In this section, we aimed to confirm presence of the well-known electrophysiological basis of fMRI signals in our intracranial EEG-fMRI dataset. Particularly, our CRP analysis suggests that fMRI and iEEG connectome patterns spatially reconfigure along largely independent trajectories. However, ample prior evidence suggests that, at the level of individual connections rather than distributed FC patterns, fMRI-FC has an electrophysiological basis readily measurable in iEEG and scalp EEG (*Pan et al., 2011*; *Wirsich et al., 2020*). To counter any apparent contradiction between our findings and this prior work, we investigated the temporal associations between time-varying fMRI-FC and iEEG-FC at each connection using the Mutual information index. Note that while iEEG-FC in our study is derived after band-pass filtering the channel-wise raw signal timecourse in the frequency band of canonical oscillations (δ to γ), the strength of cross-channel FC has a broad spectrum with strong power in the infraslow range. We investigated these temporal associations across region pairs between fMRI connectivity and iEEG connectivity extracted from HRF-convolved amplitude changes of bandlimited iEEG recordings. As shown in *Figure 3—figure supplement 2*, we found that a considerable percentage of connections show temporally associated dynamics across fMRI-FC and iEEG-FC$_{Amp}$ of all canonical frequency bands (delta through gamma: 34 ± 21%, 46 ± 19%, 47 ± 21%, 38 ± 25%, 30 ± 23%; cumulatively across all bands: 81 ± 24%). Importantly, convergence rates increased considerably when accumulated across all frequency bands. This observation speaks to the presence of *multi-frequency* association of fMRI and iEEG FC changes at the level of connections. These findings are in line with the observed cross-modal multi-frequency association at the level of connectomes.

Altogether, our findings describe two complementary aspects of FC. We confirm a readily measurable electrophysiological basis for the inherently infraslow fMRI-FC fluctuations at the connection-wise level. At the same time, we reveal additional FC phenomena that are apparent as distributed FC patterns and are temporally independent across fMRI and iEEG/EEG, thus likely capturing independent neural processes (see Discussion).

## Addressing alternative sources of contribution

Source leakage, i.e., the simultaneous detection of a brain signals from one source at several sensors, may cause spurious connectivity between electrode pairs and, therefore, distort estimation of FC matrices (*Schoffelen and Gross, 2009*). Given that intracranial EEG with macro electrodes does not suffer from source leakage (*Badier and Chauvel, 1995*; *Dubey and Ray, 2019*), our main findings cannot be driven by volume conduction effects. For the concurrent source-localized EEG-fMRI data, all results were replicated after minimizing source leakage using a state-of-the-art correction procedure, i.e., orthogonalization of source signals (*Figure 7—figure supplement 1*).

In addition to source leakage, other sources of artifacts exist that may have affected our findings. Here, we show that our findings are independent of such confounding factors or artifacts. Specifically, we quantified the association of the cross-modal similarity timecourses (as exemplified in *Figure 2B*) with: (i) distance to time-averaged FC, (ii) overall FC strength, (iii) head motion, and (iv) epileptiform activity. Each confounding factor is discussed thoroughly in the following:

### Fluctuations in static FC organization

Could the sparse and asynchronous nature of spatial convergence arise from the fact that fMRI and iEEG connectomes share a static FC organization that is expressed to *varying degrees* over time? Our above-detailed observations of shared discrete connectome states strongly speak against this scenario. Nevertheless, if the static scenario were true, the cross-modal similarity timecourse (*Figure 2B*) should scale with the similarity to the static FC organization of either modality. To test this scenario, we first calculated the correlation between frame-by-frame connectome configurations in iEEG or fMRI to the static (i.e. time-averaged) connectome of that modality (i.e. static connectome 'prominence' *timecourse*). In each frequency band, we then extracted the temporal correlation between the cross-modal similarity timecourse and the fluctuating static connectome 'prominence' timecourse of fMRI or EEG (*Figure 7—figure supplement 1*). This temporal correlation was consistently small in all comparisons (90 total cases of Subjects×Frequency bands×Modality) with the strongest positive association reaching a group-average maximum of *r*=0.07 ± 0.13 (*r*=0.14 ± 0.10 in 260 comparisons of source-localized EEG-fMRI dataset). When compared to a subject-specific chance level derived from phase-permuted cross-modal similarity timecourses (mean of 100 repetitions) no difference was observed in either dataset (for δ to γ bands, $t_8$ at most 0.54 in

intracranial EEG-fMRI, and $t_{25}$ at most 1.28 for δ to γ bands in scalp EEG-fMRI). In other words, high spatial similarity across data modalities does not temporally align with moments of particularly strong expression of the static connectome organization. These findings suggest that instances of high spatial cross-modal similarity contain contributions from shared FC features beyond the static connectome. This finding is in line with the observation of distinct connectome states at significant CRP entries.

## Limited FC strength

If the observed divergence of fMRI and iEEG FC reconfigurations were primarily driven by lack of sufficient SNR, then the strength of cross-modal FC similarity would likely depend on the strength of the signal, i.e., FC strength (for investigation of noise, see iii and iv.). In each frequency band, we assessed the temporal correlation between the frequency-specific cross-modal similarity timecourse and the frame-by-frame FC strength averaged (root mean square) overall connections of either fMRI or EEG (*Figure 7—figure supplement 1B*). The group-average temporal correlations were consistently small over all comparisons (five frequencies for cross-modal similarity ×2 modalities for FC strength ×two datasets), such that the maximum group-average value among all comparisons was $r$=0.06 ± 0.12 in the intracranial EEG-fMRI dataset, and $r$=0.14 ± 0.07 in the scalp EEG-fMRI dataset. No significant association was observed when compared to the same phase-permutation null model as described in section i (for δ to γ bands, $t_8$ at most 0.45 in intracranial EEG-fMRI, and $t_{25}$ at most 1.90 in scalp EEG-fMRI). This observation speaks against a substantial contribution of low and fluctuating signal strength to the intermittent and sparse nature of the cross-modal similarity.

## Head motion

To quantify the effects of head motion in each subject, we assessed the temporal correlation between the cross-modal similarity timecourse and the timecourse of framewise displacement (FD). Results are visualized in *Figure 7—figure supplement 1C*. Again, the group-averaged temporal correlation was consistently small, reaching a group-average maximum of $r$=0.02 ± 0.07 and $r$=0.04 ± 0.07 in the intracranial and scalp EEG-fMRI datasets, respectively. There was no significant association when compared to the same phase-permutation null model as specified in section 'i' (across bands, $t_8 \leq$ 0.21 in intracranial EEG-fMRI, and $t_{25} \leq$ 0.60 in scalp EEG-fMRI). This observation renders unlikely a substantial contribution of head motion to the intermittent and sparse nature of the cross-modal similarity.

## Epileptiform activity

In the intracranial data, electrodes corresponding to epileptic regions were marked by clinicians and excluded from our analyses. However, additional interictal activity in the healthy brain regions and their corresponding changes in connectome configurations may be a source of artifact. The generalizability of our findings to the scalp EEG-fMRI dataset of healthy subject strongly suggests the independence of our findings from epileptiform activity. Despite that, we further quantified the impact of epileptiform activity in the intracranial EEG-fMRI data. Specifically, we extracted a binary timecourse of epileptiform activity via visual inspection of the iEEG data and quantified its temporal overlap (Jaccard index) with the binarized timecourse of cross-modal similarity (timepoints of significant and non-significant cross-modal similarity marked in a binary manner). The group-averaged Jaccard index was consistently small across all frequency bands, reaching a group-average maximum of 0.06±0.05 overlap. When compared to a null model generated by randomly shifting the binarized cross-modal similarity timecourse (100 repetitions), there was no significant positive association in any of the frequency bands (one-sided paired $t$-test across subject against the subject-specific mean of null samples: $t_8$ <0.42 for δ to γ bands). This observation implies that our findings are not primarily driven by epileptiform activity (*Figure 7—figure supplement 1D*).

## **Replication in sub-second window size**

We replicated our major findings of CRP and its on-/off-diagonal ratio in the iEEG-fMRI dataset using a window length of 500 ms for FC calculations. Indeed, the data does not show a substantial difference in the on-/off-diagonal ratios of the CRP entries between the 3 s and 500 ms window lengths. Specifically, the ratio was equal to 1.02±0.07 for 500 ms window length, emphasizing absence of significant temporal convergence of the connectome dynamics (see *Figure 3—figure supplement 5*). A paired t-test between group-averaged ratios across different lags confirms a lack of significant difference between the two analyses (p=0.50). This finding further emphasizes the

genuine asynchronous nature of connectome dynamics across the neural timescales measured in fMRI and electrophysiology.

## Validation of on-/off-diagonal ratio metric

We used simulated data to validate the theoretical basis of the on-/off-diagonal ratio metric and its ability to distinguish between Scenario I and Scenario II, as depicted in *Figure 1*. Scenario III was excluded from these simulations since the absence of significant entries in the CRP under Scenario III makes the on-/off-diagonal ratio irrelevant for investigation.

We defined a random quantized signal with $N$ levels to represent the recurrent manifestation of $N$ fixed connectome states in one modality. When the second modality has an identical time course to the first modality, the resulting CRP (*Figure 3—figure supplement 6*, left subplot) shows more co-occurrences of the same states on the diagonal than off the diagonal (white entries). This aligns with expectations for Scenario I, where both spatial and temporal convergence are present. Conversely, when the state time course of the second modality is a shuffled version of the state time course of the first modality, the resulting CRP (*Figure 3—figure supplement 6*, right subplot) shows dispersed co-occurrences and diminished diagonal prominence. This outcome is in line with Scenario II, where spatial convergence exists, but temporal convergence is absent. These observed differences demonstrate how the CRP effectively captures the presence or absence of temporal alignment, successfully dissociating Scenarios I and II. We defined a random quantized signal with N levels to represent the recurrent manifestation of N fixed connectome states, in one modality. Pertaining to Scenario I, *Figure 3—figure supplement 6* (left subplot) shows the case where the second modality has the identical timecourse as the first modality. There are more co-occurrences of the same states on the diagonal than off the diagonal (white entries). This is in line with Scenario 1, where both spatial and temporal convergence are present. Pertaining to Scenario 2, *Figure 3—figure supplement 6* (right subplot) shows the case where state time course of the second modality is a shuffled version of the state time course of the first modality. As depicted, co-occurrences of the same states are more dispersed, and the diagonal prominence vanishes. The observed differences between the two cases illustrate how the CRP reflects the presence or absence of temporal alignment, dissociating scenarios 1 and 2.

To quantitatively validate this observation, we calculated the on-/off-diagonal ratio across simulations with varying N values. For Scenario 2 (shuffled version), the ratio consistently remained close to 1, indicating the absence of temporal synchronization. In contrast, Scenario 1 (non-shuffled version) produced significantly higher ratios, exceeding 1, confirming the metric's ability to capture meaningful synchrony. These results demonstrate that the simulations successfully replicate the expected relationship between the two scenarios and the CRPs, and validate the theoretical foundation of the ratio metric under the defined assumptions.

