## [Editor Report · eLife Assessment]

This **important** work uses an innovative approach to understand similarities between haemodynamic and electrophysiological activity of the human brain, and how the brain might carry out multiple functions concurrently across different brain regions by using multiple timescales. The study provides **convincing** evidence to indicate that while spatially similar functional brain networks are found in both modalities, there is a tendency for these to occur asynchronously. This work will be of interest to neurophysiological and brain imaging researchers.

---

## [Referee Report · Reviewer #1 (Public review)]

The paper proposes an interesting perspective on the spatio-temporal relationship between FC in fMRI and electrophysiology. The study found that while similar networks configurations are found in both modalities, there is a tendency for the networks to spatially converge more commonly at synchronous than asynchronous timepoints.

My confidence in the findings and their interpretation has been improved by the addition of some basic simulations. It helps give confidence in the measure being used to distinguish between scenarios.

Of course, there may be other scenarios that are problematic that are not covered by the current simulations - this highlights the difficulty of making a claim based on a heuristic measure.

That said, with the simulations included and if the caveat above is acknowledged, then I think the paper is in good shape.

---

## [Referee Report · Reviewer #2 (Public review)]

Summary:

The study investigates the brain's functional connectivity (FC) dynamics across different timescales using simultaneous recordings of intracranial EEG/source-localized EEG and fMRI. The primary research goal was to determine which of three convergence/divergence scenarios is the most likely to occur.

The results indicate that despite similar FC patterns found in different data modalities, the timepoints were not aligned, indicating spatial convergence but temporal divergence.

The researchers also found that FC patterns in different frequencies do not overlap significantly, emphasizing the multi-frequency nature of brain connectivity. Such asynchronous activity across frequency bands supports the idea of multiple connectivity states that operate independently and are organized into a multiplex system.

Strengths:

The data supporting the authors' claims are convincing and come from simultaneous recordings of fMRI and iEEG/EEG, which has been recently developed and adapted.

The analysis methods are solid and involved a novel approach to analyzing the co-occurrence of FC patterns across modalities (cross-modal recurrence plot, CRP) and robust statistics, including replication of the main results using multiple operationalizations of the functional connectome (e.g., amplitude, orthogonalized, and phase-based coupling).

In addition, the authors provided a detailed interpretation of the results, placing them in the context of recent advances and understanding of the relationships between functional connectivity and cognitive states.

The authors also did a control analysis and verified the effect of temporal window size or different functional connecvitity operationalizations. I also applaud their effort to make the analysis code open-sourced.

Comments on revisions:

The authors addressed all my concerns in the previous round of review.

---

## [Author Response]

The following is the authors’ response to the previous reviews

**Public Reviews:**

**Reviewer #1 (Public review):**
The paper proposes an interesting perspective on the spatio-temporal relationship between FC in fMRI and electrophysiology. The study found that while similar networks configurations are found in both modalities, there is a tendency for the networks to spatially converge more commonly at synchronous than asynchronous timepoints. However, my confidence in the findings and their interpretation is undermined by an incomplete justification for the expected outcomes for each of the proposed scenarios.

As detailed below, the reviewer’s comment motivated us to conduct simulations to establish the relationship between the scenarios that we seek to adjudicate and the empirical outcomes.

Main ConcernFig 1 makes sense to me conceptually, including the schematics of the trajectories, i.e.:- Scenario1. Temporally convergent, same trajectories through connectome state space- Scenario2. Temporally divergent, different trajectories through connectome state spaceHowever, based on my understanding (and apologies if I am mistaken), I am concerned that these scenarios do not necessarily translate into the schematic CRP plots shown in fig 2C, or the statements in the main text, i.e.:- For scenario1, "epochs of cross-modal spatial similarity should occur more frequently at on-diagonal (synchronous) than off-diagonal (asynchronous) entries, resulting in an on-/off-diagonal ratio larger than unity"- For scenario2, "epochs of spatial similarity could occur equally likely at on-diagonal and off-diagonal entries (ratio≈1)"Where do the authors get these statements and the schematics in fig2C from? They do not seem to be fully justified via previous literature, theory, or simulations?In particular, I am not convinced based on the evidence currently in the paper, that the ratio of off- to on-diagonal entries (and under what assumptions) is a definitive way to discriminate between scenarios 1 and 2.For example, what about the case where the same network configuration reoccurs in both modalities at multiple time points. It seems to me that you would get a CRP with entries occurring equally on the on-diagonal as on the off-diagonal, regardless of whether the dynamics are matched between the two modalities or not (i.e. regardless of scenario 1 or 2 being true).This thought experiment example might have a flaw in it, and the authors might ultimately be correct, but nonetheless a systematic justification needs to be provided for using the ratio of off- to on-diagonal entries to discriminate between scenario 1 and 2 (and under what assumptions it is valid).

Thank you for raising this important point. In response, we have now included simulation results to complement our earlier authors’ response, which provided literature references and a theoretical explanation of the on-/off-diagonal ratio metric.

In the absence of theory, the authors could use surrogate data for scenario 1 and 2. For example:a. For scenario 1, run the CRP using a single modality. E.g. feed in the EEG into the analysis as both modality 1 AND modality 2. This should provide at least one example of CRP under scenario 1 (although it does not ensure that all CRPs under this scenario will look like this, it is at least a useful sanity check).

Note: This simulation was included in the previous round of author’s responses.

b. For scenario 2, run the CRP using a single modality plus a shuffled version. E.g. feed in the EEG into the analysis as both modality 1 AND a temporally shuffled version of the EEG as modality 2. The temporal shuffling of the EEG could be done by simple splitting the data into blocks of say ~10s and then shuffling them into a new order. This should provide a version of the CRP under scenario 2 (although it does not ensure that all CRPs under this scenario will look like this, it is at least a useful sanity check)The authors have provided CRP plots for option a. It shows a CRP, as expected, consistent with scenario 1. This is a useful sanity check. However, as mentioned above, it does not ensure that all CRPs under this scenario will look like this.However, the authors have not shown a CRP as per option b. As such, there is an incomplete justification for the expected outcomes of the scenarios.Note that another option, which has not been carried out, is to use full simulations, with clearly specified assumptions, for scenario1 and 2. One way of doing this is to use a simplified (state-space) setup where you randomly simulate N spatially fixed networks that are independently switching on and off over time (i.e. "activation" is 0 or 1). Note that this would result in a N-dimensional connectome state space.Using this, you can simulate and compute the CRPs for the two scenarios:a. Scenario 1: where the simulated activation timecourses are set to be the same between both modalitiesb. Scenario 2: where the simulated activation timecourses are simulated separately for each of the modalities

We followed the reviewer’s suggestion and have now included full simulations to address the concerns regarding the theory of the on-/off-diagonal ratio metric. As recommended, we defined a random quantized signal with N levels to represent the recurrent manifestation of N fixed connectome states. This setup was used to demonstrate the relationship between the two scenarios and the CRP observations used to adjudicate between the scenarios in our paper.

The CRP matrices in Fig. S10 provide an example illustration of this simulation. In the case where the two state timeseries are identical, there are more co-occurrences of the same state (white entries) on the diagonal than off the diagonal (left subplot). This is in line with Scenario 1, where both spatial and temporal convergence are present. Conversely, in Scenario 2, where state time courses are shuffled, co-occurrences of the same states are more dispersed, and the diagonal prominence vanishes (right subplot). This difference illustrates how the CRP reflects the presence or absence of temporal alignment, dissociating scenarios 1 and 2.

To quantitively validate this observation, we calculated the on-/off-diagonal ratio across simulations with varying N values. For Scenario 2 (shuffled version), the ratio consistently remained close to 1, indicating the absence of temporal synchronization. In contrast, Scenario 1 (non-shuffled version) produced significantly higher ratios, exceeding 1, confirming the metric's ability to capture meaningful synchrony. These results demonstrate that the simulations successfully replicate the expected relationship between the two scenarios and the CRPs, and validate the theoretical foundation of the ratio metric under the defined assumptions.

Minor ConcernLeakage correction. The paper states: "To mitigate this issue, we provide results from source-localized data both with and without leakage correction (supplementary and main text, respectively)." It is great that the authors provide both. However, given that FC in EEG is almost totally dominated by spatial leakage (see Hipp paper), the main results/figures for the scalp EEG should be done using spatial leakage corrected EEG data.

Thank you. We agree that source leakage is an important consideration, which is why the current work investigated the intracranial EEG-fMRI data as a primary approach and subsequently added the scalp EEG-fMRI approach. While source leakage correction is essential for addressing spurious connectivity, it can also risk removing genuine functional connectivity that includes zero-lag relationships. We are reassured by the observation that the scalp data both without and with leakage correction confirmed the findings of the intracranial data, i.e., the presence of spatial and a lack of temporal cross-modal convergence. As such we do not believe that source leakage had a considerable impact on the specific question at hand.

**Reviewer #2 (Public review):**
Summary:The study investigates the brain's functional connectivity (FC) dynamics across different timescales using simultaneous recordings of intracranial EEG/source-localized EEG and fMRI. The primary research goal was to determine which of three convergence/divergence scenarios is the most likely to occur.The results indicate that despite similar FC patterns found in different data modalities, the timepoints were not aligned, indicating spatial convergence but temporal divergence.The researchers also found that FC patterns in different frequencies do not overlap significantly, emphasizing the multi-frequency nature of brain connectivity. Such asynchronous activity across frequency bands supports the idea of multiple connectivity states that operate independently and are organized into a multiplex system.Strengths:The data supporting the authors' claims are convincing and come from simultaneous recordings of fMRI and iEEG/EEG, which has been recently developed and adapted.The analysis methods are solid and involved a novel approach to analyzing the co-occurrence of FC patterns across modalities (cross-modal recurrence plot, CRP) and robust statistics, including replication of the main results using multiple operationalizations of the functional connectome (e.g., amplitude, orthogonalized, and phase-based coupling).In addition, the authors provided a detailed interpretation of the results, placing them in the context of recent advances and understanding of the relationships between functional connectivity and cognitive states.The authors also did a control analysis and verified the effect of temporal window size or different functional connecvitity operationalizations. I also applaud their effort to make the analysis code open-sourced.
**Recommendations for the authors:**

**Reviewer #2 (Recommendations for the authors):**
The authors answer my concerns and they are resolved.